# Plasticity in striatal dopamine release is governed by release-independent depression and the dopamine transporter

Mark D. Condon[1], Nicola J. Platt[1], Yan-Feng Zhang [1], Bradley M. Roberts[1], Michael A. Clements[1], Stefania Vietti-Michelina[1], Min-Yee Tseu[1], Katherine R. Brimblecombe [1], Sarah Threlfell [1,2], Edward O. Mann [1] & Stephanie J. Cragg [1,2]

Mesostriatal dopaminergic neurons possess extensively branched axonal arbours. Whether action potentials are converted to dopamine output in the striatum will be influenced dynamically and critically by axonal properties and mechanisms that are poorly understood. Here, we address the roles for mechanisms governing release probability and axonal activity in determining short-term plasticity of dopamine release, using fast-scan cyclic voltammetry in the ex vivo mouse striatum. We show that brief short-term facilitation and longer short term depression are only weakly dependent on the level of initial release, i.e. are release insensitive. Rather, short-term plasticity is strongly determined by mechanisms which govern axonal activation, including K$^+$-gated excitability and the dopamine transporter, particularly in the dorsal striatum. We identify the dopamine transporter as a master regulator of dopamine short-term plasticity, governing the balance between release-dependent and independent mechanisms that also show region-specific gating.

[1] Department of Physiology, Anatomy and Genetics, University of Oxford, Parks Road, Oxford OX1 3PT, UK. [2] Oxford Parkinson's Disease Centre, Oxford OX1 3PT, UK. Correspondence and requests for materials should be addressed to S.J.C. (email: Stephanie.cragg@dpag.ox.ac.uk)

D opamine (DA) release in the striatum plays key roles in action selection and behavioural reinforcement, and is dysregulated in diverse disorders including Parkinson's disease and addictions. DA neurons generate action potentials tonically at frequencies below 10 Hz and in intermittent bursts with instantaneous frequencies of up to ~40 Hz in response to salient stimuli predicting reward value or action signals[1–4]. However, the extent to which action potential activity is conveyed into striatal DA release remains unclear; DA axons are important sites for local regulation[5–7] where strong short-term plasticity can distort the relationship between activity and DA output[8,9].

Despite classical accounts of axons as high-fidelity cables, axonal neurotransmitter release is shaped by factors that regulate action potential propagation and axonal excitability such as axonal morphology and branching, the expression of ion channels and presynaptic receptors, and other diverse mechanisms that regulate vesicle pools, release probability and short-term plasticity[10]. Axons of DA neurons are remarkable: they comprise vast, extensively branched arbours[11,12] that, from a binary tree model[13], can be calculated to form ~16,000 branch points per nigrostriatal neuron. Axonal properties are therefore likely to be particularly important in governing striatal DA output. One major influence is the input from striatal cholinergic interneurons onto nicotinic receptors (nAChRs) on DA axons, which promote short-term depression (STD) of DA release[7,9,14]. Even in the absence of nAChR activation, DA release shows intrinsic short-term plasticity that ranges from short-term facilitation (STF) to STD[7,9,15,16]. The underlying mechanisms are poorly understood; existing evidence suggests that intrinsic short-term plasticity might be only weakly related to initial release[15–17] but the dominant drivers remain undefined.

Here, we delineate the roles of three types of drivers that could underlie intrinsic short-term plasticity of DA release from striatal DA axons. We examine the potential roles for: firstly, initial release; secondly, $K^+$-dependent mechanisms that will govern axonal excitability and repolarization; and thirdly, the dopamine transporter (DAT). Besides mediating DA uptake, DATs generate a depolarising transport-coupled conductance in midbrain DA neurons[18,19], and have been suggested to limit vesicular release[20–23]. Thus, DATs have the potential to govern short-term plasticity of DA release via regulation of axonal activation and/or release probability ($P_r$).

We reveal that initial release plays a limited role in short-term plasticity except on the shortest of timescales (10–25 ms), when STF operates. The prevailing STD at longer timescales (25–200 ms) is independent of the level of initial release, i.e. it is "release-insensitive". Rather, STD is strongly determined by mechanisms that influence membrane activation, particularly in dorsal striatum. Furthermore, we identify that DATs drive STF at shortest intervals and STD at longer, physiological inter-pulse intervals, effectively clamping release. We propose a region-specific hierarchy of interacting drivers of short-term plasticity, overseen by DATs, with DATs promoting release-insensitive over release-dependent mechanisms.

## Results

### Short-term plasticity in DA release is weakly release-dependent.
We assessed short-term plasticity of electrically evoked DA release in acute coronal striatal slices for pulses paired at inter-pulse intervals (IPI) of 10–200 ms in dorsolateral striatum (caudate-putamen, CPu) and nucleus accumbens core (NAc). Confounding effects of nAChRs were excluded throughout by inclusion of the antagonist DHβE (see Methods). The ratio of DA release evoked by the second pulse compared to a single pulse (paired-pulse ratio, PPR) decreased with IPI in CPu (Fig. 1a, c)

and in NAc (Fig. 1b, c). Short-term facilitation (STF, PPR > 1) was observed at the shortest IPIs (10 ms in CPu, 10–25 ms in NAc), whereas strong short-term depression (STD) (PPR < 1) was observed at longer IPIs (≥25 ms in CPu, >40 ms in NAc). STD was more pronounced in CPu than NAc (Fig. 1c, two-way ANOVA, region × IPI interaction, $F_{4,25} = 4.195$, $P = 0.010$, $n = 7$) suggesting region-specific gating of short-term plasticity.

PPR at classic fast transmitter synapses is typically inversely related to initial $P_r$: STF occurs where $P_r$ is low, and STD when $P_r$ is high[24–26]. STF at those synapses consists of several temporally overlapping processes with a range of time constants from 30 to 500 ms[27]. An inverse relationship between PPR and single pulse release of DA has been reported in CPu for IPIs of 10 ms but not 100 ms[15], suggesting that short-term plasticity of DA release might reflect initial $P_r$ at only very short intervals. We probed the relationship between PPR and the extracellular concentration of DA ($[DA]_o$) evoked by a single pulse (as a proxy for initial $P_r$) across a population of release sites with a 10-fold range in $[DA]_o$ evoked by a single pulse (1p $[DA]_o$). We found an inverse relationship between 1p $[DA]_o$ and PPR for IPI of 10 ms in CPu and NAc, and at 40 ms in NAc, but in neither region at 100 ms (Fig. 1d, linear regressions, 10 ms, CPu: $\beta = -0.79 \pm 0.14$ [95% CI: −1.08 to −0.50], $F_{1,33} = 30.9$, $P = 3.56 \times 10^{-6}$, $n = 35$; NAc: $\beta = -0.52 \pm 0.19$ [95% CI: −0.93 to −0.11], $F_{1,17} = 7.136$, $P = 0.016$, $n = 19$; 40 ms, CPu, $P = 0.152$, $n = 35$; NAc 95% CI: −0.38 to −0.001], $F_{1,17} = 4.489$, $P = 0.049$, $n = 19$; 100 ms, CPu: $P = 0.122$, $n = 35$; NAc: $P = 0.917$, $n = 19$). These data indicate that DA PPR is a function of $P_r$ at only very short intervals, when STF can also be observed. These data also suggest that short-term plasticity in DA release is governed by release-dependent mechanisms at short intervals but by mechanisms unrelated to initial release at longer intervals.

The expression of STD was not due to activation of $D_2$-autoreceptors (D2Rs). Antagonism of D2Rs with L741,626 (1 μM) did not influence peak $[DA]_o$ (Fig. 1e) or PPR at IPIs up to 200 ms in CPu or NAc (Fig. 1f, two-way ANOVA, CPu: $P = 0.420$, $n = 5$, NAc: $P = 0.2197$, $n = 6$), although it did for longer IPIs (Supplementary Fig. 1) (>200 ms, <3 s), consistent with previous findings and with no tonic $D_2$ action on DA release in slices[8,28,29]. We also ruled out an alternative hypothetical effect whereby higher DA remaining at shorter IPIs might drive more STF. We varied the intensity of the initial electrical stimulus to halve the $[DA]_o$ evoked in CPu, and showed that the $[DA]_o$ evoked by a second pulse at a fixed IPI of 25 ms and a fixed stimulus strength remained constant regardless of the initial level of $[DA]_o$ (Supplementary Fig. 2).

### $Ca^{2+}$-dependent STF and release-insensitive STD.
To test whether short-term plasticity can be shaped by $P_r$, we varied initial release by varying the extracellular concentration of $Ca^{2+}$ ($[Ca^{2+}]_o$) (1.2–3.6 mM). $[DA]_o$ evoked by 1p was dependent on $[Ca^{2+}]_o$, in CPu and NAc (Fig. 2a, b), as expected. There was a steeper relationship between $[Ca^{2+}]_o$ and 1p $[DA]_o$ in CPu than NAc (Fig. 2c, two-way ANOVA, $[Ca^{2+}]_o \times$ region interaction, $F_{2,20} = 17.25$, $P = 4.43 \times 10^{-5}$, $n = 6$), consistent with previous findings[16]. However, $[Ca^{2+}]_o$ had only modest impact on PPR, limited to very short intervals. In CPu, a trend towards an inverse relationship at 10 ms IPI did not reach significance (Fig. 2d, f, two-way ANOVA, $P = 0.604$, $n = 4$). In NAc, PPR varied inversely with $[Ca^{2+}]_o$ but at only 10 ms IPI (Fig. 2e, f, two-way ANOVA, IPI × $[Ca^{2+}]_o$ interaction, $F_{8,30} = 4.497$, $P = 0.001$, $n = 3$). Furthermore, the size of the effect on PPR in NAc versus CPu did not tally with the size of the effect on 1p $[DA]_o$; the range of 1p $[DA]_o$ was smaller in NAc than in CPu despite the larger range in short-term plasticity (Fig. 2f). The difference in short-term

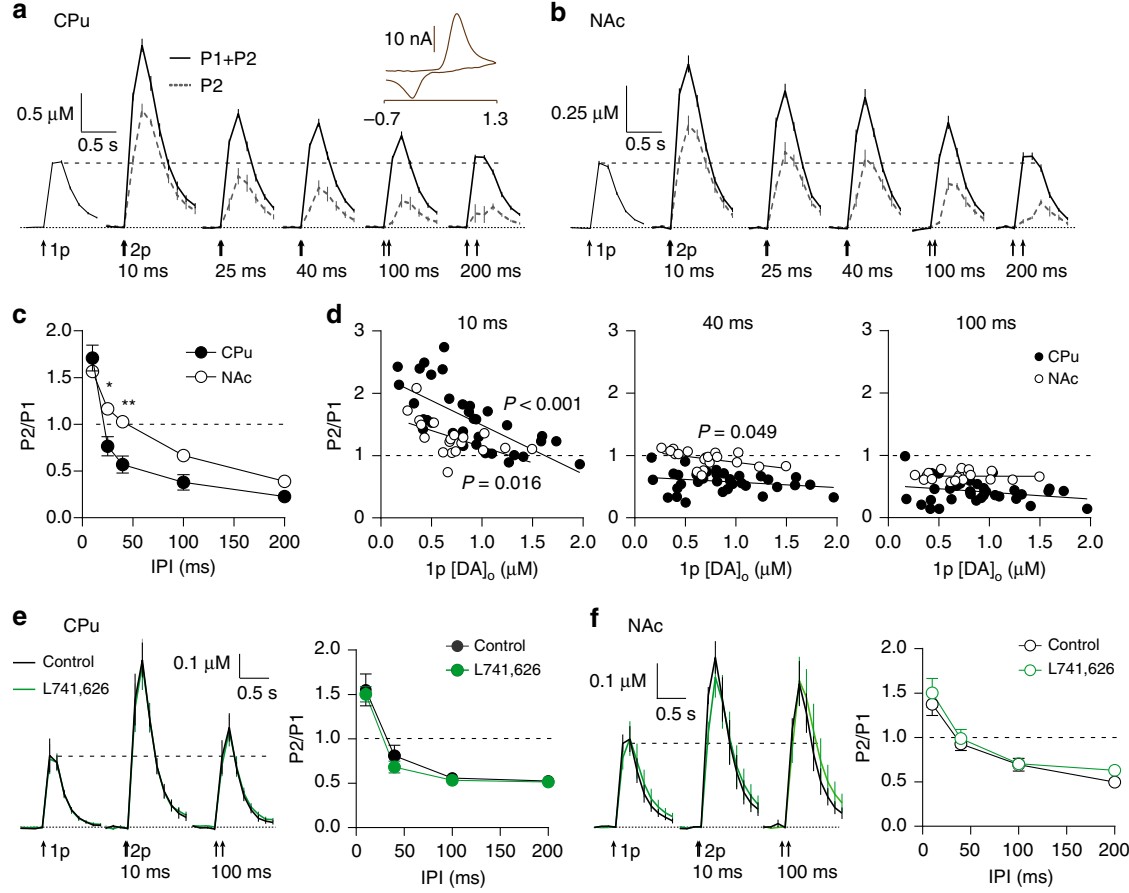

**Fig. 1** Short-term plasticity at DA release sites is region-specific and only weakly release-dependent. **a**, **b** Mean profiles of $[DA]_o$ transients elicited by single or paired electrical pulses (arrows) at IPI of 10–200 ms in CPu (**a**, $n = 4$) and NAc (**b**, $n = 3$). Dotted traces, $[DA]_o$ attributable to the second pulse, P2, in a pair ($(P1 + P2) - P1$). Inset, representative voltammogram for evoked DA release. **c** Mean P2/P1 vs IPIs in CPu (filled circles) and NAc (open circles). Where error bars are not visible they fall within the bounds of the symbol. **d** P2/P1 vs peak 1p-evoked $[DA]_o$ for IPIs of 10 ms (left), 40 ms (centre) and 100 ms (right). Each data point is 1 site in CPu (filled circles, $n = 35$) or NAc (open circles, $n = 19$). Linear fits to data show *P*-values (F-test) to indicate whether the slope is significantly > 0. **e**, **f** (Left) Mean profiles of $[DA]_o$ transients in CPu (**e**, $n = 5$) and NAc (**f**, $n = 6$) in control conditions vs L741,626 (1 μM, green). (Right) Mean P2/P1 vs IPIs in control conditions and L741,626 in CPu (**e**) or NAc (**f**). Two-way ANOVA with Bonferroni's test for post hoc comparisons: *$P < 0.05$, **$P < 0.01$. Error bars are ± SEM. Source data are provided as a Source Data file

plasticity between CPu and NAc cannot be attributed to the sensitivity of initial $P_r$ to $[Ca^{2+}]_o$ (see Fig. 2c). Together these data show that short-term plasticity of DA release is not easily explained by $Ca^{2+}$-limited mechanisms, suggesting that other mechanisms dominate that are independent of initial release.

To confirm that STD and insensitivity to $[Ca^{2+}]_o$ do not result from stimulation of an undefined input to DA axons, we used targeted optogenetic stimulation. In striatum from heterozygote $DAT^{IRES-Cre}$ mice expressing ChR2-eYFP after viral delivery, a brief light flash (2 ms, 473 nm wavelength) evoked $[DA]_o$ transients that varied with $[Ca^{2+}]_o$ (Fig. 2g) and with a steeper $[Ca^{2+}]_o$ concentration-response curve in CPu than in NAc (Fig. 2h, nonlinear regression, CPu: $R^2 = 0.91$, $n = 3$, NAc: $R^2 = 0.94$, $n = 3$, comparison of fits: $F_{4,118} = 5.94$, $P = 2 \times 10^{-4}$), as seen for electrically evoked DA release. Furthermore, at an interval of 40 ms, at which ChR2 reliably drives action potentials in DA neurons[30] and DA release[7], PPR was greater in NAc than CPu. However, even with optogenetic stimulation, there was no inverse relationship between PPR and 1p $[DA]_o$ (Fig. 2i, linear regression, CPu: $P = 0.874$, $n = 3$, NAc: $P = 0.286$, $n = 3$). These data confirm that STD does not arise from stimulation of other inputs. For subsequent experiments, we used electrical stimulation to avoid confounding effects of $Ca^{2+}$ entry through ChR2 on $P_r$ and short-term plasticity.

**Release-independent STD is controlled by K$^+$-dependent gating**. We tested whether STD was gated by axonal membrane excitability by varying $[K^+]_o$. Varying $[K^+]_o$ can alter membrane potential and repolarisation through Nernstian driving forces underlying K$^+$-mediated currents and through K$^+$-dependent inhibition of K$^+$-channel inactivation[31]. Reduced $[K^+]_o$ can thereby promote membrane hyperpolarisation and repolarisation between pulses, promoting de-inactivation of Na$^+$ channels, but can also promote inactivation of K$^+$-channels, leading to use-dependent depolarisation and enhanced Na$^+$-channel recruitment.

Varying $[K^+]_o$ (1.25–7.5 mM) did not change 1p $[DA]_o$ in CPu (Fig. 3a) or NAc (Fig. 3b) but nonetheless modulated short-term plasticity (Fig. 3a–d). In CPu, PPR varied inversely with $[K^+]_o$ across IPIs, with STD minimised at lowest $[K^+]_o$ (Fig. 3c, two-way ANOVA, $[K^+]_o \times IPI$ interaction, $F_{8,30} = 7.66$, $P = 1.53 \times 10^{-5}$, $n = 3$). In NAc, the effect of $[K^+]_o$ on PPR was less evident than in CPu, but showed an overall significance (Fig. 3d, two-way ANOVA, main effect of $[K^+]_o$, $F_{2,30} = 15.15$, $P = 2.83 \times 10^{-5}$, $n = 3$). Since we observed no effect of $[K^+]_o$ on 1p $[DA]_o$, these effects on PPR were unrelated to initial $P_r$ (Fig. 3e). These findings demonstrate that short-term plasticity of DA release can be dissociated from initial release, i.e. there is a release-insensitive short-term plasticity. Furthermore, they

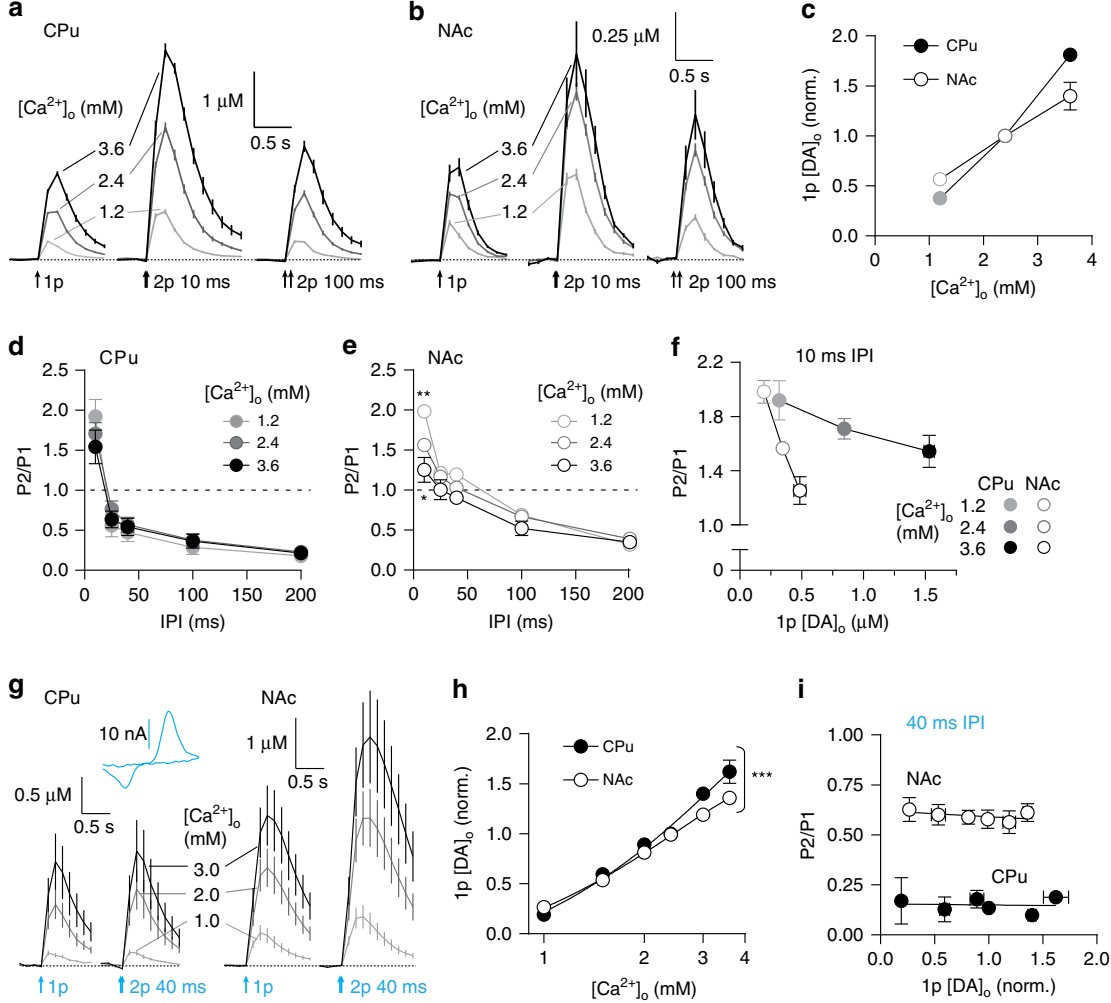

**Fig. 2** Short-term plasticity is shaped by calcium-dependent facilitation and release-independent depression. **a**, **b** Mean profiles of $[DA]_o$ transients evoked by single or paired electrical pulses in 1.2 mM (light grey), 2.4 mM (dark grey) and 3.6 mM (black) $[Ca^{2+}]_o$ in CPu (**a**, $n = 3$) and NAc (**b**, $n = 3$). **c** Mean 1p-evoked $[DA]_o$ (normalised to 2.4 mM $[Ca^{2+}]_o$) vs $[Ca^{2+}]_o$ in CPu (filled) and NAc (open circles). **d**, **e** Mean P2/P1 vs IPI in 1.2 mM (light grey), 2.4 mM (dark grey) and 3.6 mM (black) $[Ca^{2+}]_o$ in CPu (**d**, filled) and NAc (**d**, open circles). **f** Mean P2/P1 vs mean peak 1p $[DA]_o$ at 10 ms IPI in 1.2 mM (light grey), 2.4 mM (dark grey) and 3.6 mM (black) $[Ca^{2+}]_o$ in CPu (filled) and NAc (open circles). **g** Mean profiles of $[DA]_o$ transients elicited by single or paired light pulses at 40 ms IPI (473 nm, 2 ms) in 1.0 mM (light grey), 2.0 mM (dark grey) and 3.0 mM (black) $[Ca^{2+}]_o$ in CPu (left, $n = 3$) and NAc (right, $n = 3$). Inset, representative voltammogram for optogenetically evoked DA release. **h** Mean peak 1p $[DA]_o$ (normalised to 2.4 mM $[Ca^{2+}]_o$) vs $[Ca^{2+}]_o$ (log scale) evoked by light stimulation in CPu (filled) and NAc (open circles). Second-order polynomial curve fit; F-test of relative sum of squares shows curve fits are different. **i** Mean P2/P1 at 40 ms IPI vs mean peak 1p $[DA]_o$ evoked by optical stimulation in CPu (filled) and NAc (open circles). Two-way ANOVA with Bonferroni's test for post hoc comparisons except where stated otherwise: *$P < 0.05$, **$P < 0.01$, ***$P < 0.001$. Error bars are ± SEM. Source data are provided as a Source Data file

suggest that STD is governed by axonal membrane polarisation/ activation, particularly in CPu.

To identify whether the control of short-term plasticity by $[K^+]_o$ is accounted for by voltage-dependent effects, we tested whether the effects of $[K^+]_o$ on PPR could be substituted for, and prevented, by broadly blocking $K_v$ channels. We hypothesised that a $K_v$ blocker (4-aminopyridine, 4-AP) should promote STD and prevent the effects of changes in $[K^+]_o$. In CPu, broad inhibition of $K_v$ channels by 4-AP (100 μM), unlike changes in $[K^+]_o$ alone, profoundly increased 1p $[DA]_o$ (Fig. 4a, b, two-way ANOVA, main effect of 4-AP: $F_{1,3} = 33.63$, $P = 0.010$, $n = 4$), an effect consistent with action potential widening[32] and increased $Ca^{2+}$ entry. In addition, 4-AP, like high $[K^+]_o$ (see Fig. 3), but unlike high $[Ca^{2+}]_o$ (see Fig. 2), reduced PPR across IPIs, and prevented the effects of $[K^+]_o$ on PPR (Fig. 4c, d, two-way ANOVA, 4-AP × $[K^+]_o$ interaction, $F_{1,12} = 6.003$, $P = 0.031$, $n = 4$).

These findings indicate that inhibition of $K_v$ channels promotes STD, enhanced $K_v$ currents relieve STD, and that hyperpolarizing/repolarizing conditions limit STD and promote subsequent release.

In NAc, we observed no significant effect of 4-AP on 1p $[DA]_o$ (Fig. 4e, f, two-way ANOVA, $P = 0.266$, $n = 3$). However, 4-AP did decrease PPR at all IPIs, and there was no effect of $[K^+]_o$ in control conditions or in the presence of 4-AP (Fig. 4g, h). These findings verify that STD can be distinct from the effects on initial $P_r$ and identify that STD is underpinned by the gating of axonal activation.

**$K^+$-dependent gating does not alter release-dependence.** We investigated whether $[K^+]_o$-dependent STD prevents DA release from reflecting changes in $P_r$ or whether further regulatory mechanisms might be involved. We relieved STD (using low

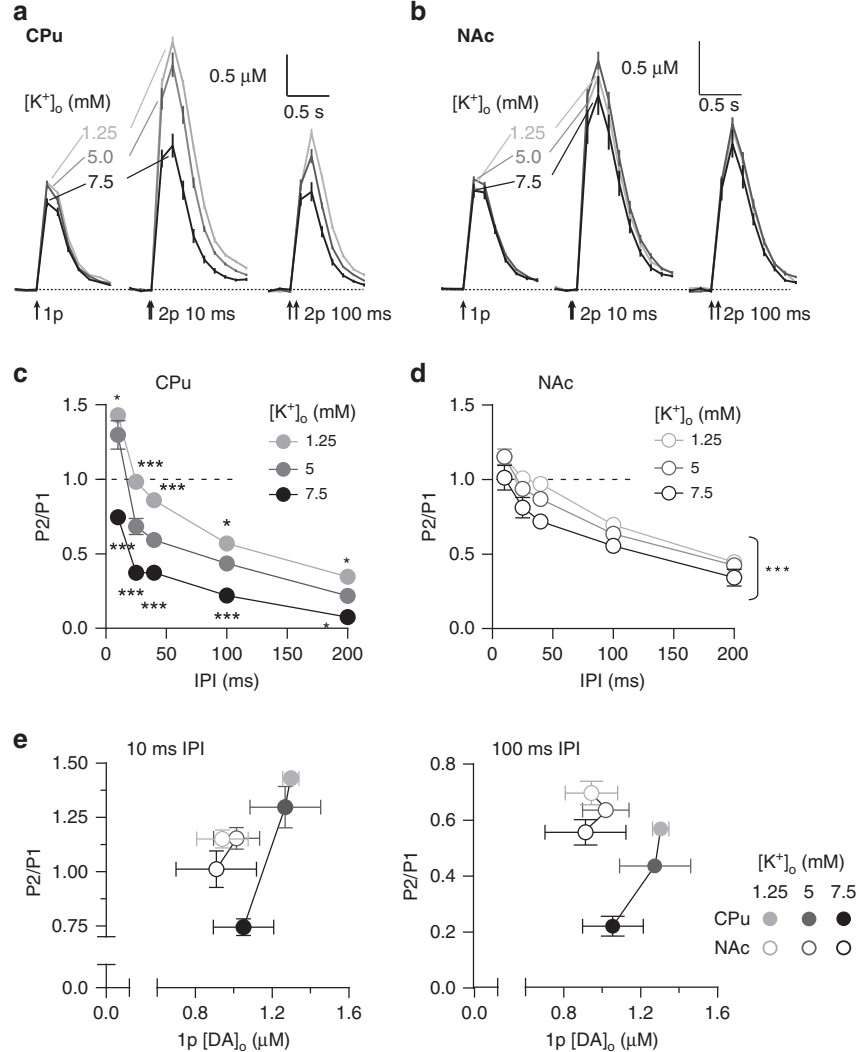

**Fig. 3** A release-insensitive mechanism of short-term depression is controlled by [K+]-dependent gating. **a**, **b** Mean profiles of [DA]o elicited by single or paired electrical pulses in 1.25 mM (light grey), 5 mM (dark grey) and 7.5 mM [K+]o (black) in CPu (**a**, $n = 3$) and NAc (**b**, $n = 3$). **c**, **d** Mean P2/P1 (± SEM) vs IPIs in 1.25 mM (light grey), 5 mM (dark grey) and 7.5 mM [K+]o (black) in CPu (**c**) and NAc (**d**). **e** Mean P2/P1 vs mean peak 1p [DA]o in 1.25 mM (light grey), 5 mM (dark grey) and 7.5 mM (black) [K+]o in CPu (filled) and NAc (open circles) at 10 ms IPI (left) and 100 ms IPI (right). Two-way ANOVA with Bonferroni's test for post hoc comparisons; *$P < 0.05$, ***$P < 0.001$. Error bars are ± SEM. Source data are provided as a Source Data file

[K+]o) to test whether we could unmask a sensitivity of short-term plasticity to changes in initial release (using variation in [Ca2+]o). In CPu, 1p [DA]o varied with [Ca2+]o in a similar manner in 1 mM [K+]o versus 5 mM [K+]o (Fig. 5a–c) and although PPRs were elevated when [K+]o was low (Fig. 5d) (as seen in Figs. 3c, 4c), there was no significant interaction between [K+]o and [Ca2+]o on PPR (Fig. 5d, two-way ANOVA, 10 ms IPI: $P = 0.086$, 100 ms IPI: $P = 0.654$, $n = 10$). In NAc, 1p [DA]o also varied with [Ca2+]o in a manner that did not depend on [K+]o (Fig. 5e–g) and there was no significant interaction between the effects of [K+]o and [Ca2+]o on PPR (Fig. 5h, two-way ANOVA, 10 ms IPI: $P = 0.963$, 100 ms IPI: $P = 0.883$, $n = 7$). By limiting STD, we did not enhance Ca2+-dependent modulation of STF or STD in either CPu or NAc. This observation suggests that an additional mechanism may operate to limit the Ca2+- and release-dependence of short-term plasticity besides Kv-regulation of axonal excitability.

**DATs regulates short-term plasticity of dopamine release.** Alongside mediating DA uptake, dopamine transporters (DATs)

have been shown to govern underlying DA release processes[20–23]. DATs also mediate electrogenic currents during DA binding and transport[18,33] that modulate the membrane potential of DA neurons in vitro[19]. We investigated the hypothesis that DATs in striatum could consequently contribute to short-term plasticity.

To avoid adaptations to release seen after DAT knockout, we used inhibitors to prevent DAT function. Monoamine uptake inhibitors cocaine (5 μM), methylphenidate (MPH, 5 μM) and DAT inhibitor nomifensine (10 μM) altered the pattern of short-term plasticity in a similar manner. They increased peak amplitude and decreased the decay rate of [DA]o transients in CPu and NAc, in keeping with DA uptake inhibition (Fig. 6a, c, g, i), and also prevented STF and relieved STD (Cocaine: Fig. 6a, b, two-way ANOVA, cocaine × IPI interaction, $F_{4,30} = 11.97$, $P = 6.24 \times 10^{-6}$, $n = 4$, MPH: Fig. 6c, d, two-way ANOVA, MPH × IPI interaction, $F_{4,40} = 5.316$, $P = 0.002$, $n = 5$; Nomifensine: Fig. 6e, f, two-way ANOVA, nomifensine × IPI interaction, $F_{4,80} = 21.12$, $P = 6.5 \times 10^{-12}$, $n = 3$). Thus, DATs are key regulators of short-term plasticity of DA release. At the shortest IPIs, DATs promote STF, whereas at longer IPIs, DATs clamp

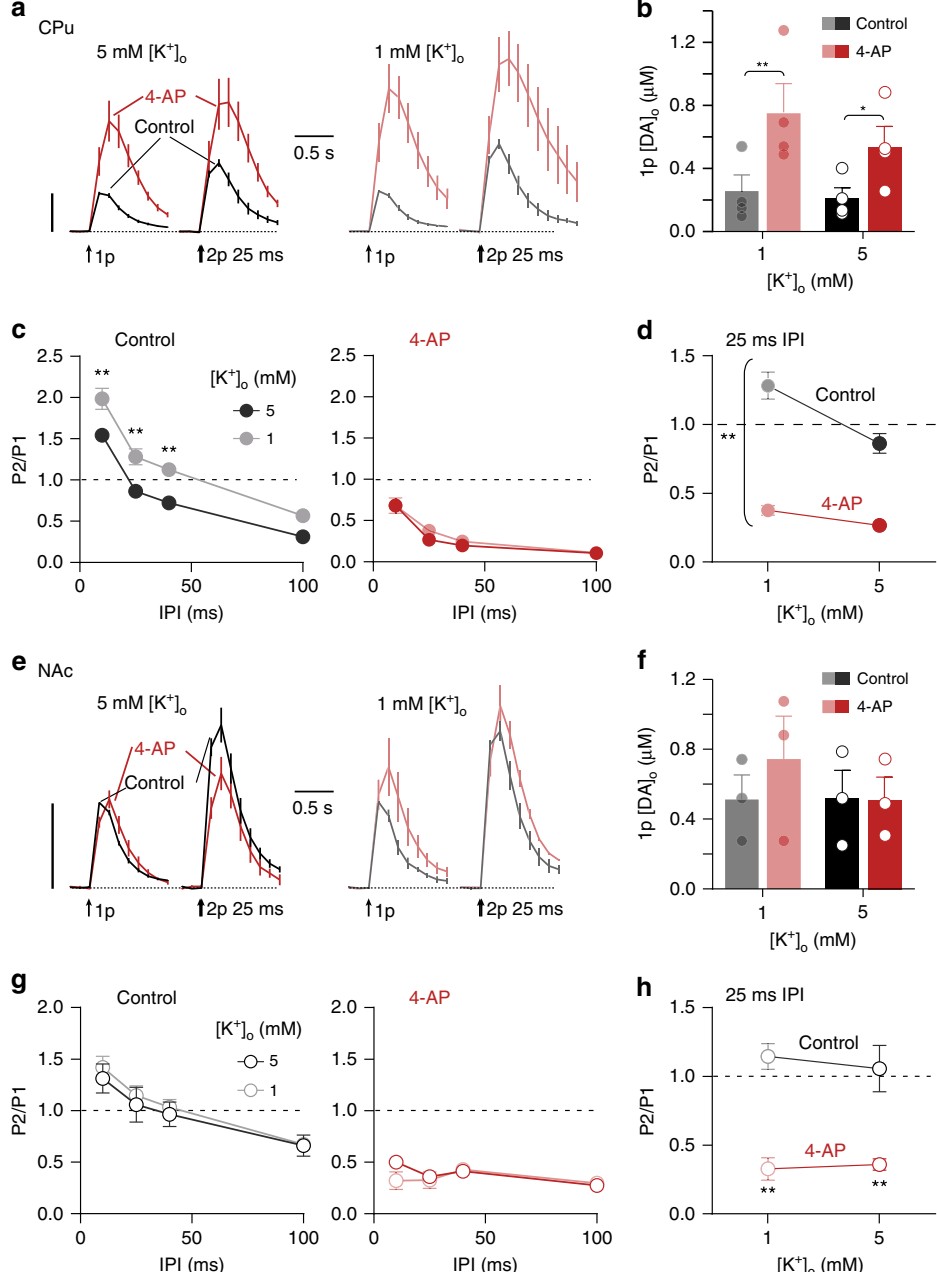

**Fig. 4** $K^+$-dependent gating of short-term depression requires axonal Kv channels. **a**, **e** Mean profiles of $[DA]_o$ transients in CPu elicited by single or paired electrical pulses in control conditions (black) or with 100 µM 4-AP (grey) in CPu (**a**, $n = 4$) and NAc (**e**, $n = 3$). Data are normalised to peak 1p-evoked $[DA]_o$ (P1) in control conditions (vertical scale bars). **b**, **f** Mean 1p $[DA]_o$ (± SEM), and individual data points in CPu in 1 mM and 5 mM $[K^+]_o$ in control conditions (black/grey) or 4-AP (reds) in CPu (**b**) or in NAc (**f**). **c**, **g** Mean P2/P1 in CPu vs IPI in 5 mM (black) and 1 mM (grey) $[K^+]_o$ in control conditions (left) and 4-AP (right) in CPu (**c**) and in NAc (**g**). **d**, **h** Mean P2/P1 in CPu at 25 ms IPI in 1 mM and 5 mM $[K^+]_o$ in control conditions or 4-AP in CPu (**d**) and in NAc (**h**). **d** Asterisks indicate interaction between $[K^+]_o$ and 4-AP. Two-way ANOVA with Bonferroni's test for post hoc comparisons; *$P < 0.05$, **$P < 0.01$. Error bars are ± SEM. Source data are provided as a Source Data file

release to promote STD in CPu. In NAc, cocaine and MPH (we did not test nomifensine) prevented STF at 10 ms IPI (Fig. 6g, h, two-way ANOVA, cocaine × IPI interaction, $F_{4,30} = 7.884$, $P = 1.82 \times 10^{-4}$, $n = 4$; MPH: Fig. 6i, j, two-way ANOVA, MPH × IPI interaction, $F_{4,20} = 3.986$, $P = 0.016$, $n = 3$), but in contrast to CPu, STD was not relieved. In NAc, DATs regulated short-term plasticity at only short inter-pulse intervals. Since it is at these intervals that short-term plasticity is related to $P_r$ (see Fig. 1), these data suggest that the effect of DAT inhibition on promoting subsequent DA release underlies its effects on short-term plasticity at short intervals in NAc.

We ruled out activation of $D_2$ receptors as contributing to DAT-mediated changes in short-term plasticity. At longer inter-pulse intervals of 2–3 s when sufficient time has elapsed for D2 receptors to be activated, the D2 antagonist L-741626 in the presence of cocaine can enhance evoked $[DA]_o$ at a second stimulus (Supplementary Fig. 3A, B, two-way ANOVA, main effect of L-741-626, $F_{1,64} = 28.96$, $P < 0.0001$, $n = 4$). However, the effect of cocaine on STF and STD at shorter intervals spanning 10–200 ms was not modified (Supplementary Fig. 3C, two-way ANOVA, main effect of L-741,626, $F_{1,20} = 2.76$, $P > 0.05$, $n = 4$). We also ruled out potential local anaesthetic actions of

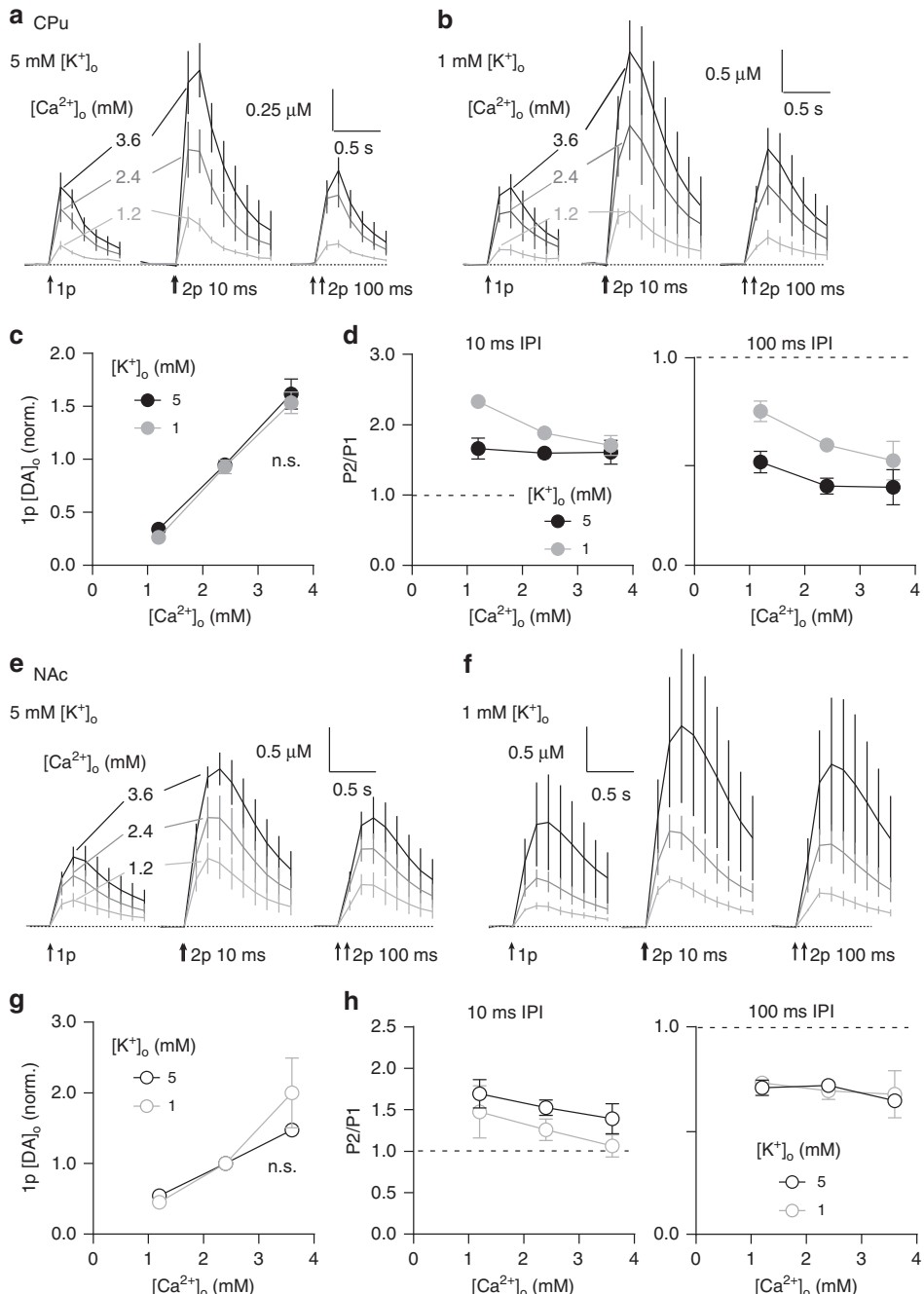

**Fig. 5** $K^+$-dependent gating does not alter release-dependence of short-term plasticity. **a**, **b**, **e**, **f** Mean profiles of $[DA]_o$ transients elicited by single or paired pulses in 1.2 mM $[Ca^{2+}]$ (light grey), 2.4 mM $[Ca^{2+}]_o$ (dark grey) and 3.6 mM $[Ca^{2+}]_o$ (black) in either 5 mM $[K^+]_o$ (**a**, **e**) or 1 mM $[K^+]_o$ (**b**, **f**) in CPu (**a**, **b**, $n = 11$) and in NAc (**e**, **f**, $n = 7$). **c**, **g** Mean peak 1p $[DA]_o$ (± SEM, normalised to 2.4 mM $[Ca^{2+}]_o$) in 5 mM $[K^+]_o$ (black) and 1 mM $[K^+]_o$ (grey) in CPu (**c**) and in NAc (**g**). **d**, **h** Mean P2/P1 at 10 ms IPI (left) and 100 ms IPI (right) in 5 mM $[K^+]_o$ (black) and 1 mM $[K^+]_o$ (grey) in CPu (**d**) and in NAc (**h**). Two-way ANOVA with Bonferroni's test for post hoc comparisons. Error bars are ± SEM. Source data are provided as a Source Data file

cocaine on voltage-gated $Na^+$ channels (VGSCs), since lidocaine (5 μM), an inhibitor of VGSCs, did not alter short-term plasticity (Supplementary Fig. 4), consistent with reports that cocaine does not inhibit VGSCs at the concentration used here[34].

We tested whether the effects of DAT inhibition on short-term plasticity were dependent on synapsin III which has been suggested to mediate the role of DATs in vesicle segregation and in limiting release[20]. However, the effects of cocaine on short-term plasticity persisted in mice lacking synapsin III (Supplementary Fig. 5), indicating a synapsin III-independent or other redundant mechanism.

**DATs limit release-dependence of short-term plasticity.** Since DATs operate a clamp on DA release in CPu that drives STD, we tested whether DATs might prevent short-term plasticity from being $Ca^{2+}$- and release-dependent. We tested whether there was a stronger relationship between $[Ca^{2+}]_o$ and PPR in CPu, in the presence of cocaine. In the absence of cocaine, there was no effect on PPR of changing $[Ca^{2+}]_o$ (Fig. 7a, b, two-way ANOVA, $P = 0.209$, $n = 4$), but in the presence of cocaine there was a significant effect of $[Ca^{2+}]_o$; PPR at IPIs of 10–40 ms was significantly elevated by lowering $[Ca^{2+}]_o$ (Fig. 7a, c–e, two-way ANOVA, $[Ca^{2+}]_o \times IPI$ interaction, $F_{3,24} = 6.137$, $P = 0.003$, $n =$

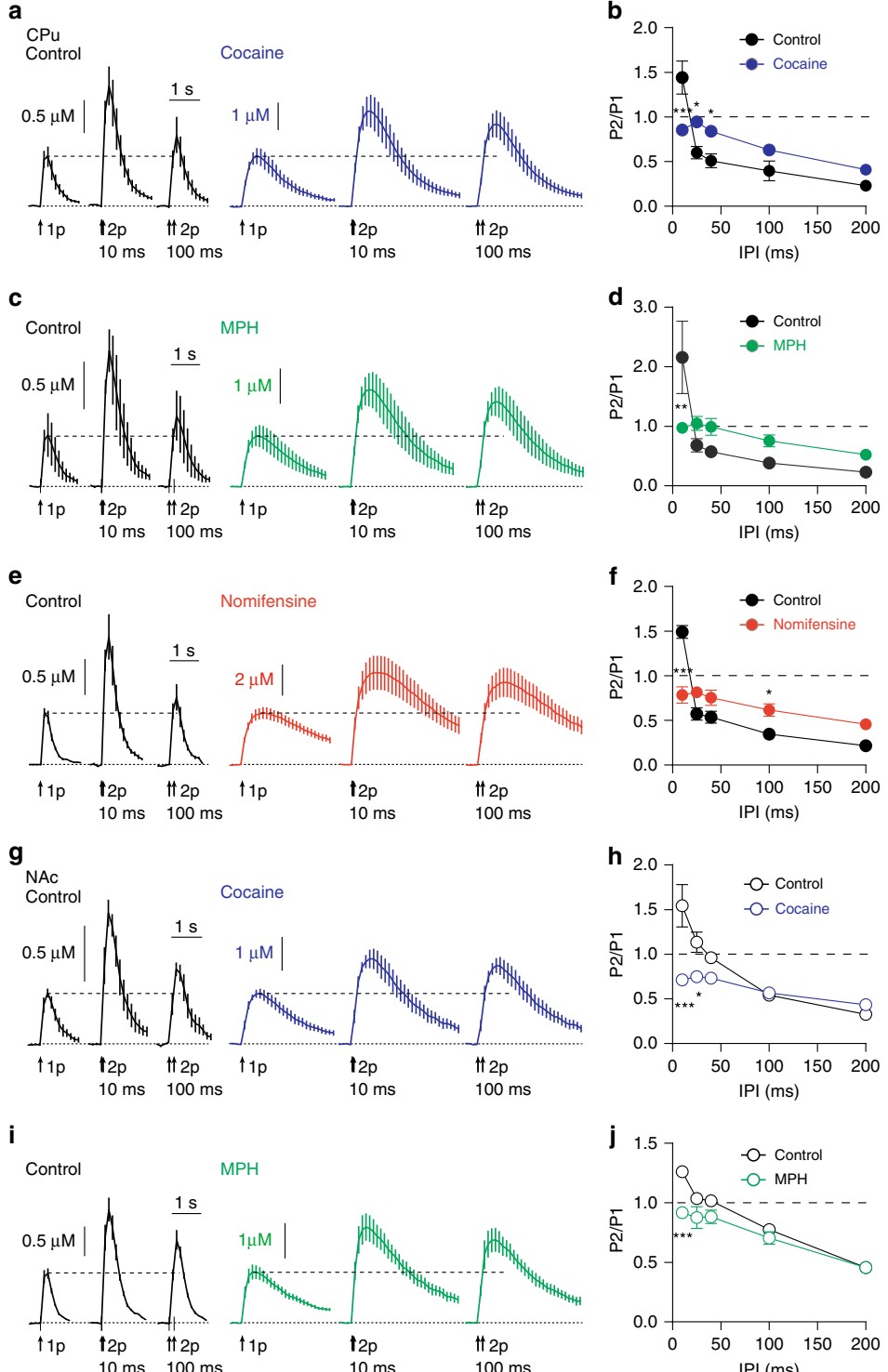

**Fig. 6** The dopamine transporter regulates short-term plasticity of dopamine release. **a**, **g** Mean profiles of [DA]$_o$ transients elicited by single or paired electrical pulses in control conditions (black) or with 5 µM cocaine (blue) in CPu (**a**, $n = 4$) or NAc (**g**, $n = 4$). **b**, **h** Mean P2/P1 against IPI in control conditions (black) and with cocaine (blue) in CPu (**b**) or NAc (**h**). **c**, **i** Mean profiles of [DA]$_o$ transients elicited by single or paired electrical pulses in control conditions (black) or with 5 µM methylphenidate (MPH, grey) in CPu (**c**, $n = 5$) or NAc (**g**, $n = 3$). **d**, **j** Mean P2/P1 vs IPI in control conditions (black) and with MPH (green) in CPu (**d**) or NAc (**j**). **e** Mean profiles of [DA]$_o$ transients elicited by single or paired electrical pulses in control conditions (black) or with 10 µM nomifensine (red) in CPu ($n = 3$). **f** Mean P2/P1 vs IPI in control conditions (black) and with nomifensine (red) in CPu. Two-way ANOVA with Bonferroni's test for post hoc comparisons; *$P < 0.05$, **$P < 0.01$, ***$P < 0.001$. Error bars are ± SEM. Source data are provided as a Source Data file

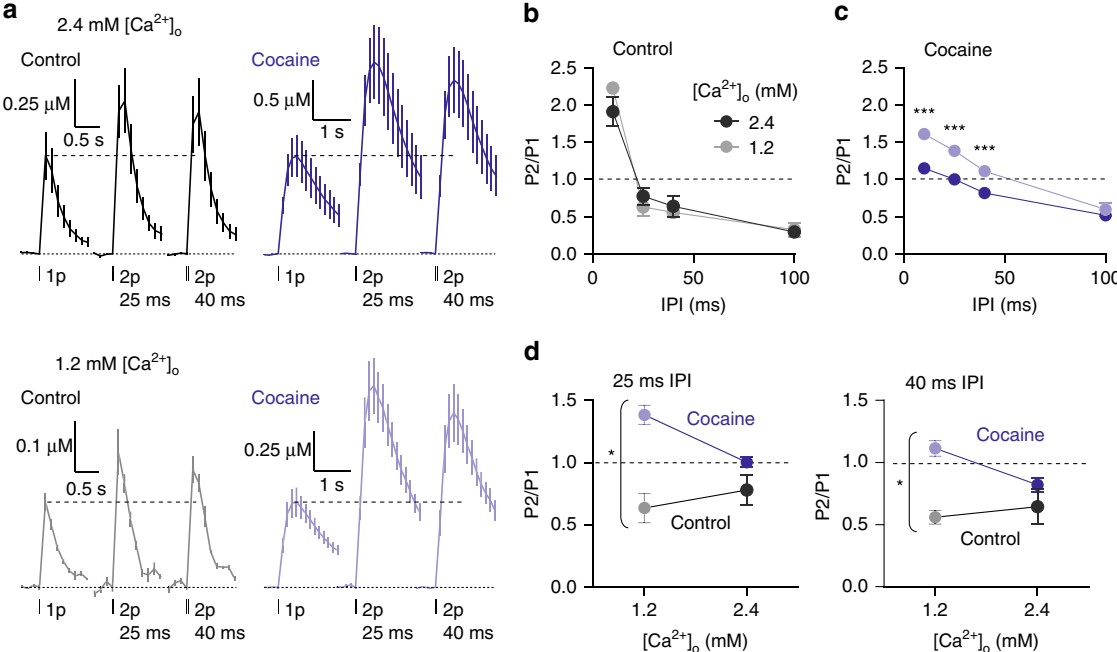

**Fig. 7** DATs limits release-dependence of short-term plasticity in CPu. **a** Mean profiles of $[DA]_o$ transients (± SEM, vertical scale normalised to 1p in 2.4 mM $[Ca^{2+}]_o$) ($n = 4$) elicited by single or paired electrical pulses in 2.4 mM (upper) and 1.2 mM $[Ca^{2+}]_o$ (lower) in control conditions (black) or 5 μM cocaine (blue). **b, c** Mean P2/P1 vs IPI in 1.2 mM (light) and 2.4 mM $[Ca^{2+}]_o$ (dark) in control conditions (**b**) or cocaine (**c**). **d** Mean P2/P1 in 1.2 mM and 2.4 mM $[Ca^{2+}]_o$ in control conditions (black-grey) or cocaine (blue) at 25 ms IPI (left) or 40 ms IPI (right). Two-way ANOVA with Bonferroni's test for post hoc comparisons; *$P < 0.05$, **$P < 0.01$, ***$P < 0.001$. Source data are provided as a Source Data file

4). There was a significantly stronger effect of $[Ca^{2+}]_o$ on PPR in the presence of cocaine than in control conditions at 25 ms (Fig. 7d; two-way ANOVA, cocaine × $[Ca^{2+}]_o$ interaction; $F_{1,12} = 7.86$; $P = 0.016$; $n = 4$) and 40 ms IPI (Fig. 7d; two-way ANOVA, cocaine × $[Ca^{2+}]_o$ interaction; $F_{1,12} = 4.72$, $P = 0.050$, $n = 4$), but not at 10 ms or 100 ms IPI. These effects were not due to a potential electrochemical change in dopamine adsorption/desorption kinetics at the electrode that might occur with a change in divalent cations, since these effects of cocaine prevailed when the reduction in $[Ca^{2+}]_o$ was compensated by substitution with $Mg^{2+}$ (Supplementary Fig. 6). Inhibition of DATs therefore relieves a limitation on the relationship between PPR and initial release, suggesting that DATs limit the release-dependence of short-term plasticity and drive STD.

**DATs maintain $K^+$-dependent gating of short-term plasticity.** The effect of DAT inhibitors on STD resembled the effect of low $[K^+]_o$. Since inhibition of DATs might prevent DAT-mediated depolarising conductances and modify axonal hyperpolarisation/repolarisation, we tested whether cocaine and low $[K^+]_o$ relieved STD via an overlapping mechanism. We explored whether cocaine precluded the effect of varying $[K^+]_o$ (1 vs 5 mM) on short-term plasticity, in CPu and NAc. In CPu, in the absence of cocaine, PPR was elevated in lower $[K^+]_o$ (Fig. 8a, b, two-way ANOVA, main effect $[K^+]_o$, $F_{1,18} = 20.47$, $P = 2.62 \times 10^{-4}$, $n = 4$) as seen in Fig. 3. Cocaine increased 1p-evoked peak $[DA]_o$ to a similar extent with varying $[K^+]_o$ (Fig. 8c) but prevented the effect on PPR (Fig. 8d, two-way ANOVA, $[K^+]_o$ × cocaine interaction, $F_{1,12} = 7.686$, $P = 0.017$, $n = 8$). The limited ability of low $[K^+]_o$ to increase PPR was not due to a ceiling effect (such as DA depletion due to increased 1p release) since cocaine also prevented the opposite effects of increased $[K^+]_o$ from decreasing PPR (Supplementary Fig. 7). In NAc, PPRs were slightly elevated in lower $[K^+]_o$ (Fig. 8e, f, two-way ANOVA, main effect $[K^+]_o$, $F_{1,12} = 6.840$, $P = 0.023$, $n = 3$), cocaine alone increased 1p-

evoked $[DA]_o$ to a similar extent with varying $[K^+]_o$ (Fig. 8g), and, as in CPu, cocaine prevented the effect of varying $[K^+]_o$ on PPR (Fig. 8f, h, two-way ANOVA, $[K^+]_o$ × cocaine interaction, $F_{1,8} = 6.756$, $P = 0.032$, $n = 3$). The effects of $[K^+]_o$ on short-term plasticity are therefore abolished by cocaine, suggesting that short-term plasticity is regulated by a pathway common to $[K^+]_o$ and DATs. DATs therefore appear to regulate short-term plasticity by limiting $Ca^{2+}$-dependent gating whilst supporting $[K^+]_o$-dependent modulation.

**DATs and $K^+$ regulate axonal activation.** Finally, to validate that $[K^+]_o$ and DATs directly modulate the activation of DA axons, through mechanisms upstream of vesicular $P_r$, we imaged axonal $Ca^{2+}$ dynamics during single and paired stimulus pulses at two IPIs, in a population of DA axons in CPu using genetically encoded calcium indicator GCaMP6f expressed in DAT-Cre: Ai95D mice (Fig. 9a). We noted firstly, in control conditions, that axonal $Ca^{2+}$ levels evoked by 2p were significantly greater than those evoked by 1p, and were greater for IPIs of 10 ms than 40 ms ($P < 0.001$, two-tailed paired $t$-test, $n = 10$) (Fig. 9b–d), which paralleled our observations for $[DA]_o$ (e.g. see Fig. 1a, c). When we increased $[K^+]_o$, $Ca^{2+}$ levels in DA axons evoked by paired pulses (normalised to 1p) were slightly but significantly decreased (Fig. 9b, c, two-way ANOVA, main effect of $[K^+]_o$, $F_{1,8} = 25.95$, $P = 0.0009$, $N = 5$ animals), consistent with reduced re-activation of DA axons at a second pulse and with the enhanced STD seen for DA release (see Fig. 3c). Furthermore, when we inhibited DATs with cocaine (5 μM), there was a significant interaction between cocaine and IPI, which increased $Ca^{2+}$ levels at IPI of 40 ms, to levels equivalent to those seen at 10 ms (Fig. 9d, e, Two-way ANOVA, interaction, $F_{1,8} = 23.19$, $P = 0.0013$, $N = 5$ animals). This interaction paralleled the impact of DAT inhibition on DA release at these IPIs (compare Fig. 6a, b). This is consistent with STD in DA release being due at least in part to limited re-activation of DA axons in the presence of DAT action.

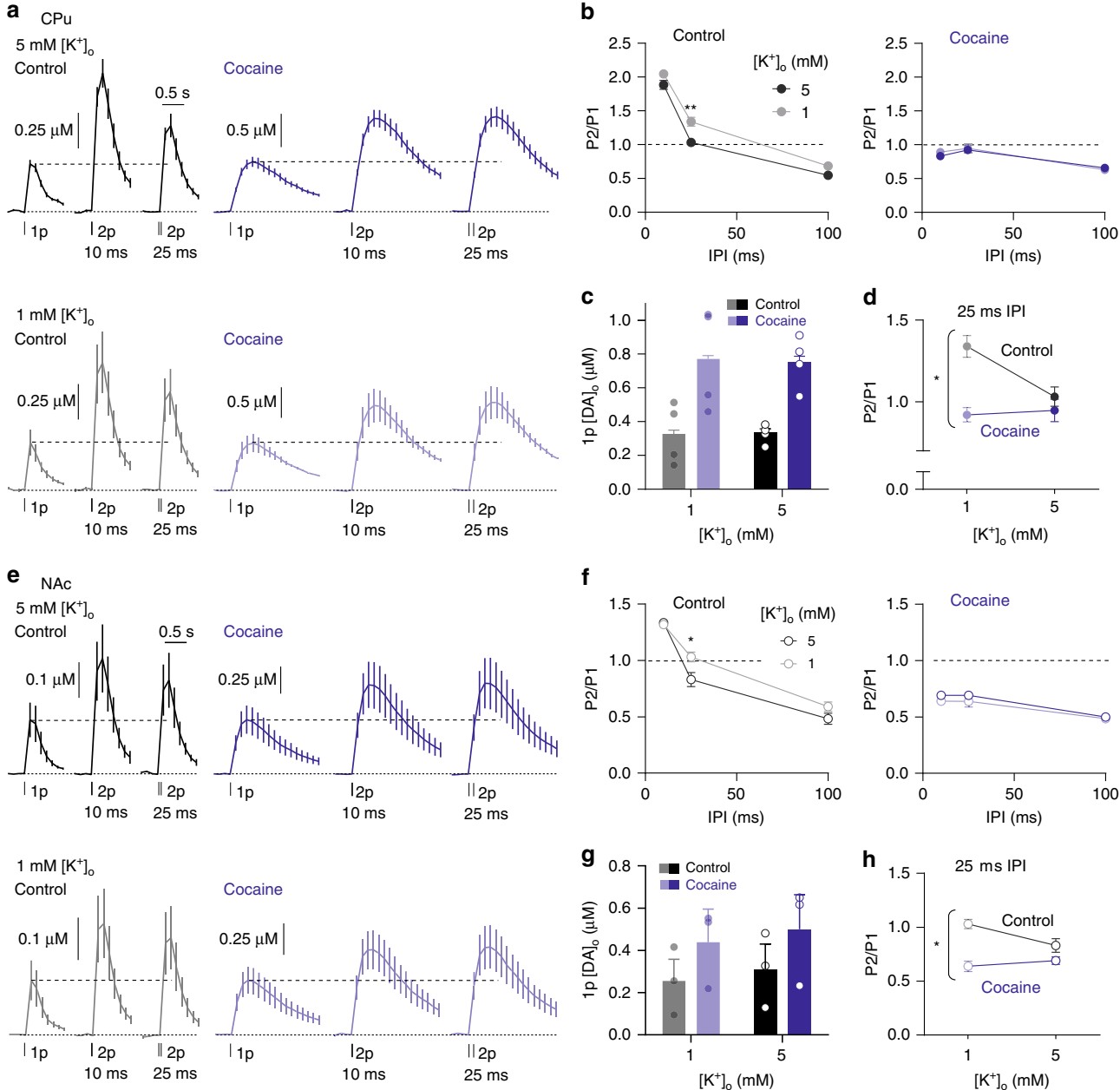

**Fig. 8** DATs and $[K^+]_o$ gate short-term plasticity through overlapping mechanisms. **a**, **e** Mean profiles of $[DA]_o$ transients elicited by single or paired electrical pulses in control conditions or cocaine in 5 mM $[K^+]_o$ (upper) and 1 mM $[K^+]_o$ (lower) in CPu (**a**, $n = 4$) and in NAc (**e**, $n = 3$). **b**, **f** Mean P2/P1 vs IPI in 5 mM $[K^+]_o$ (black) and 1 mM $[K^+]_o$ (grey) in control conditions (left) or cocaine (right) in CPu (**b**) and in NAc (**f**). **c**, **g** Mean 1p $[DA]_o$ in control conditions (black-grey) or cocaine (blue) in CPu (**c**) and in NAc (**g**). **d**, **h** Mean P2/P1 in CPu at 25 ms IPI in 1 mM and 5 mM $[K^+]_o$ in control conditions or with cocaine in CPu (**d**) and in NAc (**h**). Two-way ANOVA with Bonferroni's test for post hoc comparisons; *$P < 0.05$, **$P < 0.01$. Error bars are ± SEM. Source data are provided as a Source Data file

## Discussion

We addressed whether the mechanisms that control the short-term dynamic probability of DA release, in dorsal and ventral striatum, are governed by classic release-dependent or other release-independent mechanisms. We show that short-term plasticity is governed in only a limited manner by $Ca^{2+}$-dependent regulation of release probability, which participates in determining STF but not STD, and to greater extent in ventral than dorsal striatum. We reveal that mechanisms insensitive to the initial level of release drive strong STD, which is therefore not limited by the vesicular pool. Rather, we find that axonal excitability and DATs are major players in controlling STD, and they

dominate over $Ca^{2+}$-dependent gating. DATs appears to be master regulators that set the dynamic level of DA release and its resulting STF, and clamp release leading to release-insensitive STD, particularly in dorsal striatum. The differences seen between dorsal and ventral striatum could underlie divergent DA outputs in response to changes in action potential firing and modulatory inputs.

Short-term plasticity at fast central synapses typically demonstrates an inverse relationship between PPR and initial $P_r$; STF is observed when $P_r$ is low, and STD when $P_r$ is high[35]. Here, we show that, in NAc, an inverse relationship between STF and initial DA release could be observed for only short IPI

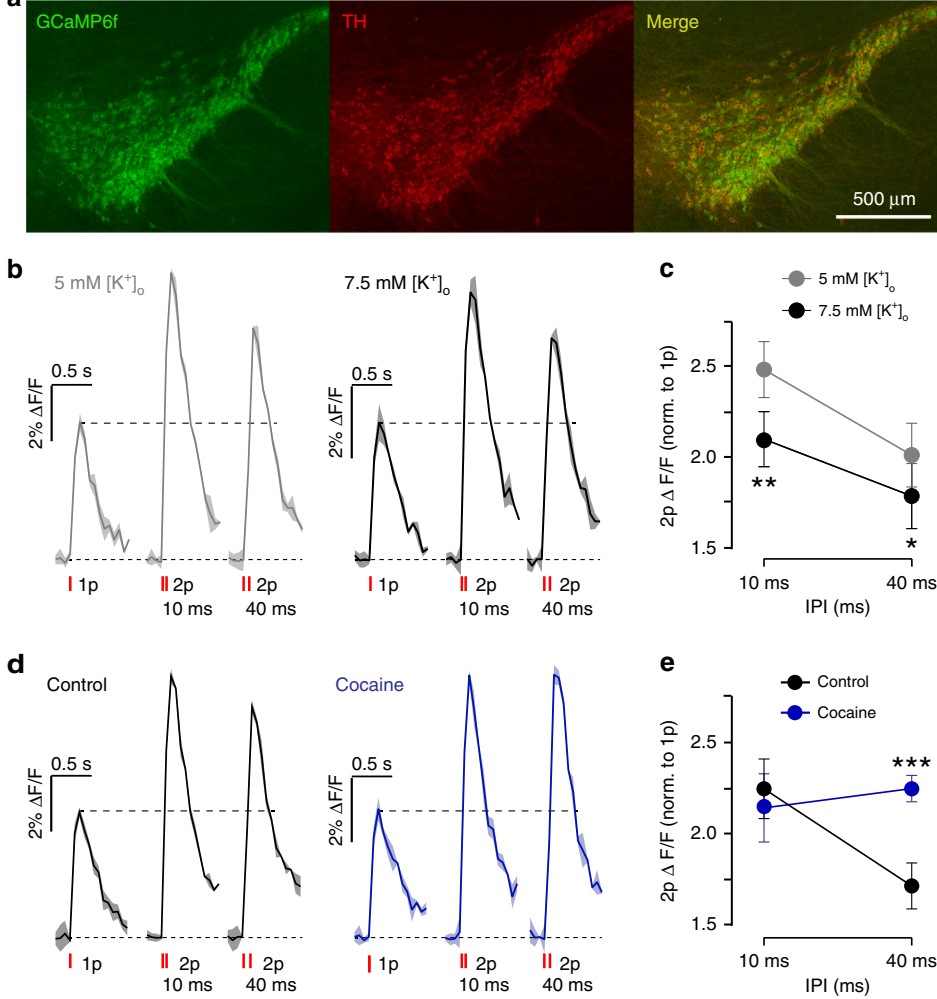

**Fig. 9** DATs and $[K^+]_o$ gate intracellular $Ca^{2+}$ dynamics during paired-pulse stimulations. **a** Images of VTA and SNc from DAT-Cre:Ai95D mice showing GCaMP6f eGFP expression (green) in TH-positive neurons (red). **b**, **d** Examples of $Ca^{2+}$-imaging responses (changes to GCaMP6f fluorescence, ΔF/F) (mean ± SEM from duplicate) in DA axon population imaged in CPu in response to single or paired electrical pulses at IPI of 10 and 40 ms in (**b**) 5 mM $[K^+]_o$ (grey) or 7.5 mM $[K^+]_o$ (black), or (**d**) in control conditions (black) or the presence of cocaine (5 µM) (blue). Data are rescaled to 1p in each condition to control for photobleaching. **c**, **e** Mean peak values for GCaMP6f ΔF/F evoked by 2 pulses vs IPI. Data are normalised to value for 1 pulse ($N = 5$ animals). Two-way ANOVA with Fishers LSD test for post hoc comparisons: *$P < 0.05$, **$P < 0.01$, ***$P < 0.001$. DHβE (1 µM) is present throughout. Error bars are ± SEM. Source data are provided as a Source Data file

corresponding to instantaneous frequencies seen during fast burst firing ($\geq 25$ Hz). STD at lower frequencies did not vary with initial release. By contrast, in CPu, there is almost no relationship between $Ca^{2+}$-limited initial release and plasticity of DA release for paired pulses. Here, other release-independent mechanisms are particularly influential. It is noteworthy that this apparent divergence from mechanisms typically operating at classic fast synapses is paralleled by differences in the molecular machinery supporting presynaptic active zones for DA release (e.g. RIM- and ELKS-dependence)[17].

STF at other synapses is usually attributed to summation of residual $Ca^{2+}$[26,36], or $Ca^{2+}$-dependent ultrafast recruitment of vesicles[37]. $Ca^{2+}$ dynamics can vary with local $Ca^{2+}$ buffering mechanisms[26]. In NAc, we saw a more pronounced $Ca^{2+}$-dependent STF than in CPu, despite a weaker relationship between $[Ca^{2+}]_o$ and initial DA release in NAc. These regional differences might correspond to differences in $Ca^{2+}$ handling. For example, the high-affinity fast $Ca^{2+}$ buffer calbindin-$D_{28k}$ occurs at 2–3-fold greater levels in DA neurons of ventral tegmental area (VTA) which innervate NAc than neurons of substantia nigra

pars compacta (SNc) which innervate CPu[38-40], and provides extra $Ca^{2+}$ buffering capacity to limit initial DA release in NAc but not CPu[41]. However, while calbindin saturation at subsequent stimuli in some other neurons promotes STF[42,43], calbindin does not apparently modify PPR in NAc[41].

At longer inter-pulse intervals (40–200 ms) corresponding to a range of physiological firing frequencies (5–25 Hz), DA release shows strong STD. STD for other transmitters can arise from depletion of readily releasable vesicles[44], but STD in DA release does not result from a limited availability of DA vesicles. Not only is a low fraction of presynaptic DA estimated to be released after stimulation[8,17,45,46], but moreover STD was not relieved by reducing initial release. $Ca^{2+}$-dependent inactivation of VGCCs by a $Ca^{2+}$ sensor has been proposed at some central synapses[47,48] but since low $[Ca^{2+}]_o$ did not relieve DA STD, this mechanism is unlikely to contribute here.

Since STD of DA release is not sensitive to the magnitude of initial release, it is a release-insensitive depression. We found particularly in CPu, that STD varied with $[K^+]_o$, as seen at some other central synapses[49]. Variation in STD with constant initial

release further demonstrates the uncoupling of STD from $P_r$. The effects of $[K^+]_o$ on DA release and on axonal $Ca^{2+}$ levels showed that factors underpinning the ability to re-activate DA axons, such as membrane polarity/excitability, upstream of local regulation of vesicular $P_r$, are critical in determining STD for DA release. A range of $K_v$ channel-types can regulate excitability and repolarisation of CNS axons[50–53]. Varying $[K^+]_o$ would be expected to change the Nernstian driving force for active $K^+$ currents, altering action potential waveform and repolarisation and therefore the degree of $Na^+$ channel inactivation, or alternatively might alter the rate of $K^+$ channel inactivation[31], leading to use-dependent changes in membrane potential which alter $Na^+$ channel recruitment. We cannot distinguish here which of these opposing mechanisms dominates to govern short-term plasticity but in either scenario, $Na^+$-channel recruitment would be altered.

Together, these findings suggest that $Ca^{2+}$-dependent mechanisms can modify amplitude of DA signals, but will not change the dynamic contrast in DA signals when DA neurons change their firing frequency, except at very highest frequencies in NAc. Conversely, mechanisms that modulate the driving forces on membrane potential, will particularly influence dynamic contrast in DA signals.

The critical role for axonal excitability in DA STD is particularly pertinent given DA axon morphology. Midbrain DA neurons form small diameter, unmyelinated, extensively arborised axons with ~$10^4$ branch points[12,54]. These morphological features will not readily favour reliable conduction of action potentials[55]. Action potential properties and propagation failure could be key contributors to DA STD. The extent of DA axonal arbour invaded could be dynamically adapted by presynaptic activity and by neuromodulatory inputs. In this regard, it is noteworthy that nAChRs regulate DA release and can drive a strong STD that limits the frequency sensitivity in DA signals[7,9]. ACh input to nAChRs might play a strategic role in shaping action potential propagation and fidelity throughout the axonal arbour. Furthermore, the stronger $K^+$-dependent gating of STD we saw here in CPu than NAc might suggest that action potential propagation might be more dynamically gated in CPu than in NAc, leading to different extents of engagement of their axonal trees. This speculation could be tested in future studies with significant implications for distinct signal processing by these neurons.

DATs in striatum regulate DA transmission through several means. DATs can curtail the extracellular summation and lifetime of $[DA]_o$ through re-uptake, and also limit the DA release process[20–23]. Correspondingly, we show that DAT function promotes STF of DA release at short intervals when short-term plasticity of DA release co-varies most strongly with initial release. Furthermore, at longer intervals, corresponding to typical firing frequencies (5–25 Hz), when release is dominated by release-independent depression, we found that DATs promote STD, particularly in CPu. DAT function therefore can therefore promote both STF and STD. These effects are akin to changing both $Ca^{2+}$-dependent $P_r$ and $K^+$-dependent excitability, but neither in isolation, and therefore indicate that DATs are limiting initial DA release probability and subsequent release through polarisation-dependent mechanisms.

The effects of DAT inhibition on STF at short IPIs seemed large given that, under control conditions, the relationship between initial release and STF was weak. This disparity is reconciled by our finding that DAT inhibition permitted $Ca^{2+}$-dependent modulation of short-term plasticity, suggesting that DATs are a critical player that controls and limits the relationship between $Ca^{2+}$ and release probability. The mechanisms are not yet known. DATs have previously been suggested to inhibit $Ca^{2+}$-dependent vesicle mobilisation via interactions with synapsins,

with synapsin-3 indicated as a potential candidate[20,21]. But our data with synapse-3 KOs do not readily support a synapsin3-dependent inhibition of vesicle recruitment by DATs. DATs are also electrogenic transporters that mediate a depolarising current[18,19,33] and can interact directly with VGCCs[56] which might influence depolarisation-dependent $Ca^{2+}$ dynamics in DA axons. However, $Ca^{2+}$ imaging in DA axons in CPu did not reveal an impact of DAT inhibition on $Ca^{2+}$ levels for paired pulses at very short IPIs where STF occurs, suggesting that the impact of DATs on STF at very short intervals seems to be downstream of $Ca^{2+}$ entry, e.g. in the mobilisation of vesicle pool or local regulation of vesicular $P_r$. There might be redundancy within the synapsin family to continue to support a synapsin-dependent mechanism in the absence of synapsin-3.

DATs, like $[K^+]_o$, also acted to promote depression at longer, physiological IPIs (40–200 ms). The overlap between the effects on STD of DATs, $[K^+]_o$ and $K_v$ channel inhibitors suggested that DATs acts through mechanisms that govern axonal activation. Using $Ca^{2+}$ imaging we validated that DATs indeed limit the ability to activate DA axons at subsequent stimuli at these intervals. Thus, DATs might act to attenuate propagation of subsequent action potentials through the DA axon arbour. Since hyperpolarisation and low $[K^+]_o$ can promote action potential renewal and propagation in some axon types[57], the potential for DATs, like high $[K^+]_o$, to depolarise membrane potential[18,19] might contribute to poor axonal re-activation in DA axons. It is of note that DATs are widely distributed throughout the length of DA axons[58], and are found at locations thought to correspond to both release-active zones and inactive zones[17]. DAT function can also be modulated by DA D2 and D3-receptors[59,60]. DATs could be ideally positioned to govern a variety of processes including action potential propagation on axons and at branches, action potential waveform at release-active zones (e.g. ref. [52]) and/or the recruitment of vesicles or VGCCs at active zones, modulated by DA. The higher density of DATs in dorsolateral striatum, underpinned by greater DAT transcript levels in SNc than VTA neurons[40], could lead to stronger limitations on action potential waveforms in CPu than NAc. Now defined, these hypotheses should be tested directly in future studies.

It is well known that DAT function can promote the frequency-dependence of DA release[15,22,61,62]. These effects are consistent with the effects of uptake on extracellular summation between stimuli and also with the role of DAT in the dynamic probability of release we have identified here. DATs might then determine both the fidelity and the spatial range of striatal DA transmission not only through their established roles in limiting DA diffusion, but also, by limiting the probability of DA release and extent of activation of the axonal arbour.

In conclusion, we propose a hierarchy of intrinsic mechanisms that control short-term plasticity of DA release. Within this hierarchy, DATs represents a 'master regulator', governing the balance between release-dependent and release-independent mechanisms that differently dominate in dorsal versus ventral striatum. In turn, DAT inhibitors such as cocaine will have profound effects on DA signalling through promoting $P_r$, relieving STD, and altering the timecourse and spatial field of DA signals.

## Methods

**Animals.** Experiments were carried out using adult male C57Bl6/J mice (Jackson Laboratories), heterozygous DAT[IRES-Cre] mice, synapsin-III knockout (S3KO) mice, or DAT-Cre:Ai95D mice on a C57Bl6/J background. For experiments using optical stimulation of DA axons, male heterozygote DAT-internal ribosome entry site (IRES)-Cre mice were bred from homozygous DAT[IRES-Cre] mice on a C57Bl6/J background (B6.SJL-Slc6a[3tm1.1(cre)Bkmn]/J, stock # 006660, Jackson Laboratories). S3KO mice were bred as described previously[63], and were kindly supplied by Professor HT Kao (Brown University). For $Ca^{2+}$ imaging experiments, male

heterozygous DAT-Cre:Ai95D mice (4–8 weeks) were bred from homozygous DAT-Cre mice (B6.SJL-Slc6a³tm1.1(cre)Bkmn/J, JAX stock number 006660) crossed with homozygous Ai95D mice (B6;129S-Gt(ROSA)26Sortm95.1(CAG-GCaMP6f)Hze/J, JAX stock number 028865). Animals were group-housed and maintained on a 12-h light/dark cycle with ad libitum access to food and water. All procedures were performed in accordance with the Animals in Scientific Procedures Act 1986 (Amended 2012) with ethical approval from the University of Oxford, and under authority of a Project Licence granted by the UK Home Office.

**Surgery**. Heterozygote DAT$^{IRES-Cre}$ mice were injected intracerebrally with a Cre-inducible recombinant AAV serotype 5 vector containing an inverted gene for channelrhodopsin-2 fused in-frame with a gene encoding enhanced yellow fluorescent protein (pAAV-double floxed-hChR2(H134R)-EYFP-WPRE-pA)[7]. Mice were placed in a stereotaxic frame under isoflurane anaesthesia and a craniotomy was made above the injection site. Injections of 1 μL virus were given either unilaterally or bilaterally in either VTA (co-ordinates from Bregma in mm: AP −3.1, ML ± 0.5, DV −4.4) or in the SNc (from Bregma in mm: AP −3.5, ML ± 1.2, DV −4.0) using a 2.5 μL 33-gauge Hamilton syringe at 0.2 μL/min with a microinjector. The syringe was left in place for 10 min following each injection, then retracted slowly. Animals were maintained for at least 3 weeks following surgery to allow virus expression in striatum.

**Slice preparation**. Mice were sacrificed by cervical dislocation and the brains removed and transferred to ice-cold HEPES-based buffer containing in mM: 120 NaCl, 20 NaHCO$_3$, 6.7 HEPES acid, 5 KCl, 3.3 HEPES salt, 2 CaCl$_2$, 2 MgSO$_4$, 1.2 KH$_2$PO$_4$, 10 glucose, saturated with 95%O$_2$/5%CO$_2$. Acute 300 μm thick coronal striatal slices, containing both dorsal striatum (CPu) and nucleus accumbens core (NAc) were prepared in ice-cold HEPES-based buffer and cut using a vibratome (VT1000S or VT1200S; Leica). Slices were kept at room temperature in HEPES-based buffer for 1 h before being transferred to the recording chamber and superfused at 1.8–2.0 ml/min with bicarbonate buffer-based artificial CSF (aCSF) containing in mM: 124.3 NaCl, 26 NaHCO$_3$, 3.8 KCl, 2.4 CaCl$_2$, 1.3 MgSO$_4$, 1.2 KH$_2$PO$_4$, 10 glucose, saturated with 95% O$_2$/5% CO$_2$, at 31–33 °C. Recording medium also contained dihydro-β-erythroidine (DHβE, 1 μM). Slices were allowed to equilibrate for 30 min prior to recording.

**Voltammetry and stimulation**. Evoked extracellular DA concentration ([DA]$_o$) was measured using FCV at carbon-fibre microelectrodes (fibre diameter 7–10 μm, tip length 50–100 μm) implanted to a constant depth of 100 μm. A triangular voltage waveform was scanned across the microelectrode (−700 to + 1300 mV and back vs Ag/AgCl reference, scan rate 800 V/s) using a Millar voltammeter (Julian Millar, Barts and the London School of Medicine and Dentistry), with a sweep frequency of 8 Hz. This sampling rate is sufficient to capture the rising and falling phase of the DA transients; faster sampling rates do not change the data interpretations (not illustrated). Evoked currents were confirmed as DA by comparison of the voltammogram with that produced during calibration with applied DA in aCSF (oxidation peak + 500–600 mV and reduction peak −200 mV). Currents at the oxidation peak potential were measured from the baseline of each voltammogram and plotted against time to provide profiles of [DA]$_o$ versus time. Electrodes responded linearly to [DA]$_o$ over the concentration range detected. Electrodes were calibrated after use in 2 μM DA in each experimental solution used, including all solutions where [Ca$^{2+}$] was varied, since electrode sensitivity varies with divalent ion concentration[64]. Calibration solutions were made up immediately before use from stock solution of 2.5 mM DA in 0.1 M HClO$_4$ stored at 4 °C. Electrode sensitivities ranged from 5 to 25 nA/μM.

For experiments using electrical stimulation, DA release was evoked using a surface bipolar concentric Pt/Ir electrode (25 μm diameter, FHC) placed ~100 μm from the recording electrode. Stimulation pulses of 200 μs duration were applied at 0.6 mA unless described otherwise. All experiments were conducted in the presence of nAChR antagonist DHβE (1 μM) to prevent confounding effects of striatal ACh. Striatal nAChRs operate a profound control over striatal DA release: ACh released by striatal stimulation directly drives axonal DA release[7,65] and limits dependence of DA release on presynaptic frequency[7,9] which can mask mechanisms operating presynaptically on DA axons. Also, inclusion of nAChR antagonists during local stimulation makes evoked DA release sensitive to frequency of stimulation[9] as seen in vivo after midbrain or medial forebrain bundle stimulation[61,66]. For optogenetic stimulation, DA release was evoked in striata from DAT$^{IRES-Cre}$ mice conditionally expressing ChR2 in DA axons, using 2 ms full-field illumination from an LED system emitting light at 470 nm wavelength (OptoLED, Cairn Research)[7]. The LED system illuminated an area of 2.2 mm diameter. The current delivered by the LED power supply was set to produce a perimaximal light intensity (i.e. the minimum light intensity able to evoke maximum [DA]$_o$ following a single 2 ms pulse). Since the perimaximal light intensity is dependent on the level of ChR2 expression, which varies between animals, the appropriate current was determined at the beginning of each experiment. Electrical or optical stimulations were delivered every 2.5 min, after which dopamine release was reproducible. Before acquisition of experimental data, peak evoked [DA]$_o$ levels were allowed to reach this reproducible stable level.

**DA release study design and analyses**. The term release probability P$_r$ used for DA here is a composite measure of synaptic P$_r$ (a function of vesicular P$_r$ and the size of the pool of vesicles) and the number of release sites and fibres recruited by the stimulus. Short-term plasticity in DA release was explored by applying alternating single pulses (1p) or paired pulses (2p) with inter-pulse intervals (IPIs) of 10–200 ms in pseudorandom order and in triplicate at each recording site. IPIs of 40–200 ms fall within the range commonly observed during burst firing in DA neurons in vivo. IPIs of 10–25 ms have been observed during burst firing in rat DA neurons[2] but are particularly useful to interrogate short-term facilitation which occurs on this timescale[9,15].

We calculated paired-pulse ratio (PPR) as a measure of short-term plasticity. We define PPR as the ratio P2/P1, where P1 is peak [DA]$_o$ detected following 1p stimulus and P2 is the peak [DA]$_o$ attributable to the second stimulation only. P2 was determined by subtracting the entire [DA]$_o$ transient including decay phase after a single pulse from the summated paired-pulse response, and we therefore account for summation and decay resulting from uptake kinetics. Any enhanced spillover resulting from uptake inhibition will occur for single and paired pulses and should therefore be controlled for. We cannot control for variable fibre recruitment for different stimuli, but can control for variable modulation by other local inputs: PPR on the timecourses explored here is not differently modified at different intervals by D2 dopamine receptors (see Results), or by GABA receptors (Lopes and Cragg, unpublished observations). We use the term "release-dependent" plasticity to indicate a relationship between PPR and P1 [DA]$_o$. "Release-insensitive" or "release-independent" refers to a PPR that varies independently of P1, or does not vary with P1.

Calcium concentration ([Ca$^{2+}$]$_o$) in aCSF was varied by varying [CaCl$_2$]. In some experiments where stated, a reduction in CaCl$_2$ was substituted with MgCl$_2$. However, Ca$^{2+}$/Mg$^{2+}$ substitution was not routine because other effects of [Mg$^{2+}$]$_o$ can confound interpretation. Changes in [Mg$^{2+}$] have been shown to alter Ca$^{2+}$ currents through VGCCs[67,68], alter G-protein-coupled receptor function, including dopamine D2 receptors[69], change the affinity of calcium-binding protein calbindin for Ca$^{2+}$[70], trigger complexing of Ca$^{2+}$ by ATP[71], and alter the sensitivity of FCV electrodes to DA[64], and we find also that Ca$^{2+}$/Mg$^{2+}$ substitution reduces DA release to levels below those of reducing Ca$^{2+}$ alone (Supplementary Fig. 6D).

Where extracellular potassium concentration ([K$^+$]$_o$) was varied (1.0–7.5 mM), changes in osmolarity were compensated by varying [Na$^+$]$_o$. The Na$^+$- and K$^+$-containing salts present in aCSF were varied as follows (in mM): In experiments with 1 mM [K$^+$]$_o$, salts were 127.1 NaCl, 1.0 KCl, 0 KH$_2$PO$_4$, 1.2 NaH$_2$PO$_4$; For 1.25 mM [K$^+$]$_o$ salts were 128.0 NaCl, 0.05 KCl, 1.2 KH$_2$PO$_4$; For 7.5 mM [K$^+$]$_o$, salts were 121.8 NaCl, 6.3 KCl and 1.2 KH$_2$PO$_4$. Recordings in control conditions were collected before and after each experimental manipulation at each recording site. Slices were equilibrated in each solution for at least 20 min before data were included for analysis. Experiments were carried out in either the dorsal half of the caudate-putamen (CPu) (dorsolateral and dorsomedial striatum) or nucleus accumbens core (NAc) (within 200 μm of the anterior commissure), one site per slice. Release is TTX-sensitive[7]. Manipulations of [Ca$^{2+}$]$_o$ or [K$^+$]$_o$ did not change DA decay rates.

Data were acquired and analysed using Axoscope 10.2, 10.5 (Molecular Devices) or Strathclyde Whole Cell Programme (University of Strathclyde, Glasgow, UK) and locally written scripts in Visual Basic for Applications (Microsoft). Data are expressed as mean ± standard error (SEM) and n = number of animals (the number of different biological replicates). Each experiment was performed at a single recording site in one brain slice. Within a given experiment at each single recording site, technical replicates were usually obtained in at least triplicate before averaging to obtain the value for that experiment at that recording site. Statistical comparisons were carried out using GraphPad Prism 6, 7 and 8 (GraphPad Software Inc.) using ANOVAs, post hoc two-tailed t-tests with n = number of animals. Data were tested for normality before using parametric tests.

**Ca$^{2+}$ imaging**. An Olympus BX51Wl microscope equipped with a CAIRN Research OptoLED Lite system, Prime Scientific CMOS (sCMOS) Camera (Teledyne Photometrics), and a ×40/0.8 NA water-objective (Olympus) was used for wide-field fluorescence imaging of GCaMP6f in dopaminergic axons in dorsolateral CPu in ex vivo slices in response to single and paired electrical stimulus pulses. Images were acquired at 16.6 Hz frame rate every 2.5 min using Micro-Manager 1.4, with stimulation and recording synchronised using custom-written procedures in Igor Pro 6 (WaveMetrics) and an ITC-18 A/D board (Instrutech). Image files were analysed with Matlab R2017b and Fiji 1.5. We extracted fluorescence intensity from the region of interest (ROI) and from an equal background area where there was no GCaMP6f expression (on the stimulating electrode). After background subtraction, the Ca$^{2+}$ transients were bleach-corrected by fitting an exponential curve function through both the baseline (2 s prior to stimulation) and the last 1 s in a 7.2 s recording window. Data are expressed as ΔF/F where F is the fitted curve. The order of single and paired stimulation pulses was alternated and equally distributed, and data were collected in duplicate before and after a change in extracellular experimental condition. Data are expressed as mean ± standard error of the mean (SEM) for each stimulus condition. N value is the number of animals. Statistical analyses used GraphPad Prism 7.03, for two-way ANOVA with post hoc

*t*-tests. All experiments were conducted in the presence of DHβE (1 μM). GCaMP6f responses were quantified as the mean ΔF/F value.

**Immunocytochemistry**. We verified the specificity of GCaMP6f expression to dopaminergic structures by comparing direct eGFP fluorescence to immunoreactivity to tyrosine hydroxylase (TH-ir). Acute slices of midbrain and striatum were fixed overnight at 4 °C in 4% paraformaldehyde dissolved in PBS, then stored in PBS. After resectioning to 40 μm, free-floating sections were washed in PBS 5 × 5 min and incubated in 0.5% Triton X-100, 10% normal goat serum and 10% foetal bovine serum for 30 min. Slices were subsequently incubated overnight with 1:2000 primary (rabbit anti-TH; Sigma) antibody dissolved in PBS containing 0.5% Triton X-100, 1% normal goat serum and 1% foetal bovine serum. Sections were then washed with PBS 5 × 5 min and incubated for 2 h at room temperature with 1:1000 secondary (DyLight 594 goat anti-rabbit; Jackson) antibody dissolved in PBS containing 0.5% Triton X-100, 1% normal goat serum and 1% foetal bovine serum. Sections were washed with PBS and mounted on gelled slides with Vectashield mounting medium (Vector Labs) and imaged using a Zeiss LSM880 (confocal) running Zen black version 2.3, at 20×, N.A. 0.8. Maximum intensity projection from a z-stack of height 30 μm was captured individually and the stack of the pictures were compressed. TH (red) was captured at 638–759 nm with 633 nm excitation light. GCaMP (green) was excited with 488 nm and captured at 493–630 nm.

**Drugs**. DHβE and lidocaine were obtained from Tocris Biosciences or Ascent Scientific. Cocaine hydrochloride and nomifensine maleate were obtained from Sigma. Methylphenidate (Ritalin™) was obtained from Novartis. All drugs were dissolved in either de-ionised water, aqueous acid (nomifensine maleate) or ethanol (lidocaine) to make stock aliquots at 1000–10,000x final concentration and stored at −20 °C prior to use. DA uptake inhibitors were used at standard concentrations that have detectable outcomes on uptake kinetics concentrations but are lower than those that lead to run-down of release and/or have non-selective effects as local anaesthetics or at nAChRs[34,72,73].

**Reporting summary**. Further information on research design is available in the Nature Research Reporting Summary linked to this article.

## Data availability
The authors declare that all data sets generated and analysed during this study are available within this paper and its supplementary files. The source data underlying all figures and supplementary figures are provided as a Source Data file.

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

## Acknowledgements

Funding sources: MRC Doctoral Training Grants (to M.D.C., N.J.P.), BBSRC Doctoral Training Grants (to M.A.C., S.V.M.), Clarendon Fund Award (to B.M.R.), MRC grant (MR/K013866/1 to S.J.C.), Monument Trust Parkinson's UK Discovery Award (J-1403), Parkinson's UK (G-1305 to S.J.C.) and Christ Church Oxford.

## Author contributions

M.D.C. designed and performed experiments, analysed and interpreted the data, and co-wrote the paper. N.J.P., M.A.C. and Y.-F.Z. designed and performed experiments, and analysed and interpreted the data. B.M.R., S.V.M. and M.Y.T. performed experiments and analysed data. K.R.B., S.T., E.O.M. and S.J.C. supervised the work. S.J.C. designed the experiments, interpreted the data, and co-wrote the paper.

## Additional information

**Competing interests:** The authors declare no competing interests.

