## [Transparent Peer Review File · Nature Communications]

Reviewers' comments:

Reviewer #1 (Remarks to the Author):

This manuscript by Condon et al. examines the determinants of short-term plasticity of dopamine release in striatum. The authors show that paired pulse depression (PPD) occurs largely independently of the probability of dopamine release and external Ca^{2+} concentration. This differs substantially from most glutamatergic synapses where PPR and release probability are inversely related. Additionally, the authors show that blocking dopamine transporters with cocaine and other blockers reveals a more Ca-dependent short-term plasticity suggesting that DAT clamps the Ca dependence of the paired pulse ratio. Lastly, the authors find that the release-independent PPD can also be modulated by changes in extracellular K^+ . From these experiments, they authors conclude that the PPF and PPD observed may be produced by mechanisms that involve changes in axonal excitability.

The short-term plasticity findings reported here (eg. Ca-independent PPD, involvement of DAT in PPF) are interesting. The experimental data are all very high quality and the study is well-executed. Beyond these phenomenological observations, however, this study has a major shortcoming which is that although the authors very heavily suggest axonal mechanisms throughout the manuscript, the actual mechanisms explaining these phenomena are either lacking or are largely asserted. For example, the authors show that blocking DAT reduces both paired pulse facilitation (for short interstimulus intervals) and paired pulse depression (for longer intervals). They conclude that DAT prevents STP from being Ca^{2+} - and release-dependent but how this happens is not determined. The authors state that this effect may involve "polarization dependent mechanisms" but provide no data directly testing this assertion. In addition, the authors' assert that axonal mechanisms are involved in the short-term depression observed but never directly test this idea. Certainly, past work in cerebellum has shown that low terminal excitability contributes to depression at Purkinje to cerebellar nucleus synapses (Kawaguchi and Sakaba, 2015), but this type of depression occurs only during high frequency stimulation trains of 50- 100 Hz. In dopamine neurons, however, paired-pulse depression of 0.2 occurs even at very long interstimulus interval of 200ms – 1s. Even for very small and branchy axons (see Radivojevic et al, 2017), it is difficult to believe that such a dramatic loss of axonal excitability, increased spike failures, etc. would ever occur after only two stimuli at 1-5 Hz. With the lack of any clear mechanisms, therefore, I believe that this study would be appropriate for a more specialized journal.

Comments:

1. The authors' main conclusion is that short-term depression is dependent on regulation of axonal excitability but they fail to experimentally test this idea. The authors begin to explore this question by blocking axonal K-currents with 4-AP but the interpretation of the result is unclear. They find that 4-AP application increases depression which may be expected. However, whether increasing PPD with 4-AP reveals much about the mechanism underlying the original PPD observed under control conditions is unclear. Similarly, the authors show that changes in the extracellular potassium levels affect dopamine release. However, this is a nonspecific manipulation that influences dopamine release through a variety of mechanisms that may or may not be relevant to the mechanisms underlying the severe PPD observed in control dopaminergic neuron axons.

2. The authors show the interesting result that activation of DAT seems to promote facilitation at very high frequencies but causes depression at lower frequencies (cocaine block reduces the early PPF and late PPD). Again, they hypothesize that the electrogenic current produced by DAT or interactions of DAT with K channels may result in the observed effects. But without direct experiments, it is difficult to say whether the electrogenic current makes a significant impact.

3. The authors use L741,626 to block D2-receptors, showing that it has little effect on PPR. In other studies, however, sulpiride produces a substantially more complete block of D2-autoreceptor

mediated effects (eg, see Martel et al, 2011). The authors should justify their use of L741,626 rather than sulpiride. This is important as they use the lack of effect by L741,626 to argue that D2-receptors do little to shape STP in the presence of cocaine (Figure S1), which to me would be very surprising.

4. The authors use carbon fiber measurements of dopamine release as a proxy for release probability, but how well this measurement can be used as a consistent reflection of Pr between samples has not been verified. For example, the signals that are obtained during carbon fiber experiments reflect spillover of dopamine into the extracellular solution. Because of this, these signals may vary dramatically from recording sites, to slice depth, number of axons being stimulated. These measurements are probably valid for relative manipulations within the same recording. However, it is difficult to know how or whether comparisons between slices and between animals can be accurately performed. Further justification of this approach will be more helpful to the reader.

Reviewer #2 (Remarks to the Author):

In this ms Condon et al. investigated the role of DAT, Ca²⁺, K⁺, and basal DA concentration on short-term plasticity at DA release in slices of the NAc and dorsal striatum of mice. Short-term depression (STD) or short-term facilitation (STF) were observed when the DA release (measured by FSCV) was evoked by 1 or 2 pulses of electric stimulation. The effects these variables on dopamine release were tested by varying the inter potential intervals and the concentration of Ca²⁺ or K⁺, and DAT-blockers. Optogenetics was used to confirm the specificity of DA fibers stimulations. Overall experiments were well designed, data seems solid and the collected data are coherent with the proposed hypotheses. However, below I suggest doing an additional critical experiment to test one of the main hypotheses, ask for some clarifications, and make mild suggestions.

Main point:

1. The results presented in the ms are coherent with the hypothesis that “short-term plasticity (whether STD or STF) is weakly release-dependent”. This study showed that at short inter potential intervals STF prevailed, while STD prevailed at longer inter potential intervals. Another possible interpretation for this finding is that this happened because higher concentrations of dopamine at the time of the second pulse causes STF while lower concentrations of DA at the time of the second pulse causes STD – the concentration of dopamine after the first pulse decreases exponentially with time.

I ask the authors to disambiguate these two hypotheses by varying the current applied in the first pulse from 100 uA to 1000 uA, while maintaining the current applied in the second pulse constant (600 uA).

Minor points:

2. In Methods is informed that “n” is the number of animals and that the measures of DA release were done in triplicates. Were the number cases per group used in the stats 3 or 9?

3. Using the STP as the abbreviation for short-term plasticity is confusing because STP is more usually used as the abbreviation for short-term potentiation. In some parts of the text (i.e. in the legend of Fig. 2) I had the impression that STP has been used with the meaning of potentiation (facilitation) instead of plasticity.

4. I need to make a list of abbreviations to understand the text. I suggest the authors to add such a list into the ms.

5. Page 3, lines 58, 59: define shortest and longer timescales.

6. It is not easy to understand the meaning of the colors and symbols used to identify each group by reading the Figure legend text. I suggest printing this information in the figure itself. It is also difficult to differentiate among light grey, dark grey and black. Different colors would make it

easier to the readers.

Claudio Da Cunha

Reviewer #3 (Remarks to the Author):

Condon and co-workers present the novel finding that the dopamine transporter (DAT) plays an important role in regulating short-term plasticity (STP) of DA signaling, by controlling the balance between release-dependent and independent mechanisms. Using fast scan cyclic voltammetry to measure electrically (or optically) stimulated DA release in brain slices, they find differences in gating of DA release between dorsal striatum and nucleus accumbens core, which could be relevant to the differing role these brain regions play in important DA signaling based behaviors. These novel results contribute significantly to increasing understanding of mechanisms contributing to release probability in the extensively branched axonal arbors of DA neurons. The manuscript is well written, well organized, and statistical analyses are appropriate. There are a few concerns, mostly minor.

1. D2 autoreceptors are known to regulate activity of DAT (e.g. Dickenson et al., 1999, PMID:9886065; Lee et al., 2007, PMID:17380124). It is surprising therefore that experiments antagonizing D2 receptors (Figure 1) did not reveal any slowing in DA clearance kinetics. The authors should discuss reasons for this apparent discrepancy.
2. It seems important to include a set of control experiments showing effects of the nAChR antagonist, DH β E, on experimental endpoints. Dr. Cragg and her group have published extensively on nAChR modulation of DA neurotransmission, so even citing these previous works, with a brief description of this modulation would be sufficient.
3. Aside from nAChRs, the D3 receptor has also been shown to regulate DAT activity (Zapata et al., 2007, PMID:17923483). The authors should consider potential D3 receptor involvement in their results.
4. Rationale for concentrations of cocaine, methylphenidate and nomifensine should be provided.
5. Experiments in Supplementary Figure S4 are confusing. In the text on page 10, it is stated that "the reduction in $[Ca^{2+}]_o$ was compensated by equimolar substitution with Mg^{2+} ", but the figure shows "addition" of Mg^{2+} . Please clarify. The authors should also comment on the variance in P2/P1 in controls at 10 ms between data shown in Figure S4.B and Figure 7.B Given the relatively small SEMs, it appears there are differences here that should be discussed.
6. Under the surgical methods section, how long was the microinjector left in place after injection of virus? Details of how the accuracy of injection site was evaluated should be included. An image showing eYFP expression, and some quantitation of transduction efficiency should be included in supplementary materials.
7. It would be useful to include some representative voltammograms in either Figure 1, or in supplementary materials.
8. Similarly, results of calibrations of electrodes should be reported in supplementary materials. How did r^2 values compare between pre- and post-electrode calibrations?
9. Please clarify if recording sites were always targeted to the same area in CPu and NAc (pages 19-20).

10. In Figure 1 caption, please indicate more clearly what the arrows are indicating. Also please clarify that where error bars are not visible they fall within the bounds of the symbol.

11. In Figure 4, reference to the colors is confusing. It seems that in line 3 of the caption *grey* should be *red*, likewise in second to last line.

12. Is it possible that compensation in synapsin III knockout mice could mask a possible role for this protein on the effects of cocaine on STP? Brief discussion of this possibility and alternative approaches should be included.

Minor corrections

Page 3, 5 lines from bottom, the word "rather" should be deleted.

Page 4, first line of second paragraph, "release probability (Pr)" has already been previously defined.

[DA]_o should be defined when first used in text.

Page 8, third line of paragraph one, "relieving" should be "relieved".

Page 11, last line of first paragraph, "raise" should be "rise"; 5 lines from bottom, should "vesicle pool" be "vesicular pool"?

Page 13, 12 lines from top, "...but might will alter..." should be "...but might well alter...".

Page 14, first line of second paragraph, delete second "together" in "Together, these findings together..."

Page 15, 10 lines from bottom, "...In any event, It is...", i in It should be lower case.

Page 19, top line, "IPIs of 10 to 25 ms IPIs..." delete second IPI.

Check references for consistency in style, e.g. in most places journal names are abbreviated, but in others they are written in full.

Why are the x-axes different in Figure 2.C and Figure 2.H? Also, it is not entirely clear why there are 6 different Ca²⁺ concentrations shown in 2.H, and only 3 in 2.C. Presumably more concentrations were tested in 2.H. This was not clearly indicated.

The x-axes in Figure 3.E should be made the same.

Figures 4.A and 4.E are missing the vertical scale bar.

Control data in Figures 6.A, C, E, G and I are missing the horizontal scale bar, and Figures 6.D, F, H and J are missing the y-axis title.

Control data in Figures 8.A and E are missing the horizontal scale bar.

Data in Figure S4.A are missing scale bars.

Control data in Figure S5.A are missing horizontal scale bars.

REPLIES TO REVIEWERS

Overall comments: We thank all of the reviewers for their constructive and insightful comments. In response, we have added new data and figures that test our main conclusions, and reinforce and advance our original findings. We have addressed all of the reviewers' comments below, and revised the manuscript throughout to produce a significantly improved manuscript. Our revisions are yellow-highlighted within the revised manuscript.

Reviewer #1 (Remarks to the Author):

This manuscript by Condon et al. examines the determinants of short-term plasticity of dopamine release in striatum. The authors show that paired pulse depression (PPD) occurs largely independently of the probability of dopamine release and external Ca²⁺ concentration. This differs substantially from most glutamatergic synapses where PPR and release probability are inversely related. Additionally, the authors show that blocking dopamine transporters with cocaine and other blockers reveals a more Ca-dependent short-term plasticity suggesting that DAT clamps the Ca dependence of the paired pulse ratio. Lastly, the authors find that the release-independent PPD can also be modulated by changes in extracellular K⁺. From these experiments, they authors conclude that the PPF and PPD observed may be produced by mechanisms that involve changes in axonal excitability.

The short-term plasticity findings reported here (eg. Ca-independent PPD, involvement of DAT in PPF) are interesting. The experimental data are all very high quality and the study is well-executed. Beyond these phenomenological observations, however, this study has a major shortcoming which is that although the authors very heavily suggest axonal mechanisms throughout the manuscript, the actual mechanisms explaining these phenomena are either lacking or are largely asserted. For example, the authors show that blocking DAT reduces both paired pulse facilitation (for short interstimulus intervals) and paired pulse depression (for longer intervals). They conclude that DAT prevents STP from being Ca²⁺- and release-dependent but how this happens is not determined. The authors state that this effect may involve "polarization dependent mechanisms" but provide no data directly testing this assertion. In addition, the authors' assert that axonal mechanisms are involved in the short-term depression observed but never directly test this idea. Certainly, past work in cerebellum has shown that low terminal excitability contributes to depression at Purkinje to cerebellar nucleus synapses (Kawaguchi and Sakaba, 2015), but this type of depression occurs only during high frequency stimulation trains of 50- 100 Hz. In dopamine neurons, however, paired-pulse depression of 0.2 occurs even at very long interstimulus interval of 200ms – 1s. Even for very small and branchy axons (see Radivojevic et al, 2017), it is difficult to believe that such a dramatic loss of axonal excitability, increased spike failures, etc. would ever occur after only two stimuli at 1-5 Hz. With the lack of any clear mechanisms, therefore, I believe that this study would be appropriate for a more specialized journal.

1. The authors' main conclusion is that short-term depression is dependent on regulation of axonal excitability but they fail to experimentally test this idea. The authors begin to explore this question by blocking axonal K-currents with 4-AP but the interpretation of the result is unclear. They find that 4-AP application increases depression which may be expected. However, whether increasing PPD with 4-AP reveals much about the mechanism underlying the original PPD observed under control conditions is unclear. Similarly, the authors show that changes in the extracellular potassium levels affect dopamine release. However, this is a nonspecific manipulation that influences dopamine release through a variety of mechanisms that may or may not be relevant to the mechanisms underlying the severe PPD observed in control dopaminergic neuron axons.

RESPONSE: We appreciate that our manipulations of $[K^+]_o$ and broad-spectrum Kv blocker 4-AP are relatively nonspecific means to manipulate K⁺ currents and voltage-gated K channels. We used these approaches as tools to prove that short-term plasticity in DA release can be uncoupled from initial release, and that STD can

be driven and relieved by mechanisms that would be expected to change axonal membrane activity/excitability. We have now conducted key experiments using Ca^{2+} imaging of DA axons in striatum of DAT-Cre:Ai95D mice to test directly whether axonal activation/excitability is modified by key manipulations and can therefore underpin short-term dynamics in DA release. We manipulated $[\text{K}^+]_o$ (and DATs, see point 2 below) and imaged GCaMP6f Ca^{2+} transients in DA axons in CPu during paired pulses at two interpulse intervals. We find that a change in $[\text{K}^+]_o$ modifies the ability to activate DA axons at subsequent stimuli, consistent with the changes to short-term plasticity in DA release and also with axonal activation/excitability being a critical factor. We have added a **new figure (Figure 9A-C)**, Results and Methods, and have revised the Conclusions and Discussion throughout in the light of these new confirmatory findings. We have also more carefully re-worded the concept of axonal “excitability” by referring more to axonal “activation”, which we now show directly.

2. The authors show the interesting result that activation of DAT seems to promote facilitation at very high frequencies but causes depression at lower frequencies (cocaine block reduces the early PPF and late PPD). Again, they hypothesize that the electrogenic current produced by DAT or interactions of DAT with K channels may result in the observed effects. But without direct experiments, it is difficult to say whether the electrogenic current makes a significant impact.

RESPONSE: Again, to address this comment, we conducted new experiments with Ca^{2+} imaging of striatum from DAT-Cre:Ai95D mice to test directly whether DATs modify the excitability of DA axons. We inhibited DATs using cocaine and imaged GCaMP6f Ca^{2+} transients in DA axons in CPu during paired pulses at two key interpulse intervals. We found that cocaine promoted the re-activation of DA axons at the interval when cocaine also relieves short-term depression of DA release. These findings confirm that DATs limit axonal activation. We have added these data to a **new Figure 9D,E**, along with corresponding Results and Discussion throughout.

3. The authors use L741,626 to block D2-receptors, showing that it has little effect on PPR. In other studies, however, sulpiride produces a substantially more complete block of D2-autoreceptor mediated effects (eg, see Martel et al, 2011). The authors should justify their use of L741,626 rather than sulpiride. This is important as they use the lack of effect by L741,626 to argue that D2-receptors do little to shape STP in the presence of cocaine (Figure S1), which to me would be very surprising.

RESPONSE: We are glad to have the chance to clarify. L-741626 is selective for D2 receptors, rather than D2/D3 for sulpiride, and hence is preferable for testing for D2-specific effects. Moreover, L-741626, just like other D2 antagonists, does indeed impact on paired-pulse release. This is observed at intervals greater than those used in core experiments here: D2 autoreceptor action has been well documented to occur in a time window of between ~500 ms and 2 seconds after initial release (Schmitz et al 2002 *J Neurosci* PMID 12223553; Phillips et al 2002 *Synapse* PMID 11842442). We have now corroborated an effect of L-741626 on this timescale in a **new Supplementary Figure S1**, showing a robust effect of L-741626 for interpulse intervals of 1-2 seconds that match previous observations of D2 receptor control. The original data for cocaine with L-741626 that was in previous Fig S1 is now to be found in a figure renumbered as Supplementary Figure S3.

4. The authors use carbon fiber measurements of dopamine release as a proxy for release probability, but how well this measurement can be used as a consistent reflection of Pr between samples has not been verified. For example, the signals that are obtained during carbon fiber experiments reflect spillover of dopamine into the extracellular solution. Because of this, these signals may vary dramatically from recording sites, to slice depth, number of axons being stimulated. These measurements are probably valid for relative manipulations within the same recording. However, it is difficult to know how or whether comparisons between slices and between animals can be accurately performed. Further justification of this approach will be more helpful to the reader.

RESPONSE: We are glad to have the chance to clarify. Carbon fibers have diameters of 7-10 μm , and tip lengths of 50-100 μm , and therefore sample extracellular DA from tens to hundreds of DA release sites. We implant to a constant depth of 100 μm in all experiments. Therefore, measurements of DA release in good experimental preparations show good consistency from slice to slice, are consistent at a given site, and readily reveal regional differences e.g. CPU > NAc. Electrodes can be calibrated after release to identify the concentrations released and can be used for more than one experiment. Thus, with good experimental design, comparisons between experiments, slices and animals can accurately be performed. We have added to the Methods that recordings are made at a standardised depth in all recordings.

Reviewer #2 (Remarks to the Author):

In this ms Condon et al. investigated the role of DAT, Ca²⁺, K⁺, and basal DA concentration on short-term plasticity at DA release in slices of the NAc and dorsal striatum of mice. Short-term depression (STD) or short-term facilitation (STF) were observed when the DA release (measured by FSCV) was evoked by 1 or 2 pulses of electric stimulation. The effects these variables on dopamine release were tested by varying the inter potential intervals and the concentration of Ca²⁺ or K⁺, and DAT-blockers. Optogenetics was used to confirm the specificity of DA fibers stimulations. Overall experiments were well designed, data seems solid and the collected data are coherent with the proposed hypotheses. However, below I suggest doing an additional critical experiment to test one of the main hypotheses, ask for some clarifications, and make mild suggestions.

Main point:

1. The results presented in the ms are coherent with the hypothesis that “short-term plasticity (whether STD or STF) is weakly release-dependent”. This study showed that at short inter potential intervals STF prevailed, while STD prevailed at longer inter potential intervals. Another possible interpretation for this finding is that this happened because higher concentrations of dopamine at the time of the second pulse causes STF while lower concentrations of DA at the time of the second pulse causes STD – the concentration of dopamine after the first pulse decreases exponentially with time. I ask the authors to disambiguate these two hypotheses by varying the current applied in the first pulse from 100 μA to 1000 μA , while maintaining the current applied in the second pulse constant (600 μA).

RESPONSE: We thanks the reviewer for this interesting suggestion. We have now performed experiments that confirm that the $[\text{DA}]_o$ remaining at P1 does not determine release at P2. We varied the intensity of the initial electrical stimulus to halve the $[\text{DA}]_o$ evoked, and then used a fixed stimulus strength (600 μA) at the second stimulus at a fixed IPI of 25 ms. We find that the same $[\text{DA}]_o$ are evoked at P2 regardless of initial level of $[\text{DA}]_o$ evoked by P1. We have now added this data in a new **Supplementary Figure S2**.

Minor points:

2. In Methods is informed that “n” is the number of animals and that the measures of DA release were done in triplicates. Were the number cases per group used in the stats 3 or 9?

RESPONSE: The numbers used in the statistical analyses were the number of separate experiments (biological replicates), which was typically the same as the number of animals. For the Reviewer’s example, this would have been n = 3. We have clarified within the Methods section.

3. Using the STP as the abbreviation for short-term plasticity is confusing because STP is more usually used as the abbreviation for short-term potentiation. In some parts of the text (i.e. in the legend of Fig. 2) I had the impression that STP has been used with the meaning of potentiation (facilitation) instead of plasticity.

RESPONSE: We agree that this acronym can be confusing. We have now removed the STP abbreviation and replaced with the longhand expression “short-term plasticity” throughout.

4. I need to make a list of abbreviations to understand the text. I suggest the authors to add such a list into the ms.

RESPONSE: We have taken care to minimise the number of abbreviations used in the revised manuscript. For example, besides replacing STP, we have replaced “AP” with “action potential” which avoids some potential confusion with the drug 4-AP. We have ensured others are defined clearly. We hope this helps.

5. Page 3, lines 58, 59: define shortest and longer timescales.

RESPONSE: We have now added this helpful clarification to these lines.

6. It is not easy to understand the meaning of the colors and symbols used to identify each group by reading the Figure legend text. I suggest printing this information in the figure itself. It is also difficult to differentiate among light grey, dark grey and black. Different colors would make it easier to the readers.

RESPONSE: We have added extra data labels to key panels in several figures, and increased the contrast in grays to help.

Reviewer #3 (Remarks to the Author):

Condon and co-workers present the novel finding that the dopamine transporter (DAT) plays an important role in regulating short-term plasticity (STP) of DA signaling, by controlling the balance between release-dependent and independent mechanisms. Using fast scan cyclic voltammetry to measure electrically (or optically) stimulated DA release in brain slices, they find differences in gating of DA release between dorsal striatum and nucleus accumbens core, which could be relevant to the differing role these brain regions play in important DA signaling based behaviors. These novel results contribute significantly to increasing understanding of mechanisms contributing to release probability in the extensively branched axonal arbors of DA neurons. The manuscript is well written, well organized, and statistical analyses are appropriate. There are a few concerns, mostly minor.

1. D2 autoreceptors are known to regulate activity of DAT (e.g. Dickenson et al., 1999, PMID:9886065; Lee et al., 2007, PMID:17380124). It is surprising therefore that experiments antagonizing D2 receptors (Figure 1) did not reveal any slowing in DA clearance kinetics. The authors should discuss reasons for this apparent discrepancy.

RESPONSE: The reviewer raises an interesting previous finding that D2 receptors can under some paradigms regulate activity of the DAT. There are conflicting observations in the field. D2 knockout in mice (Dickenson et al 1999) and HEK cells can lead to reduced DA uptake (Lee et al 2007) while other studies show conversely that D2 knockout mice have elevated DA uptake (Schmitz et al 2002 PMID 12223553) and that D2 agonists reduce DATs (Kimmel et al 2001 PMID 11408534). The impact on DAT function likely depends on whether acute or chronic adaptations are involved and the extent of D2 receptor activation. On the timecourse of our experiments, which involve brief dopamine release events and acute drug applications over just tens of minutes, D2 receptor activation and any adaptations in DAT will be kept to a minimum, and small changes in DA uptake rates are unlikely to have a detectable impact on the falling phases of our subsecond transients signals for endogenous DA evoked by single pulses. See also response to point 3 regarding D3 receptors.

2. It seems important to include a set of control experiments showing effects of the nAChR antagonist, DH β E, on experimental endpoints. Dr. Cragg and her group have published extensively on nAChR modulation of DA neurotransmission, so even citing these previous works, with a brief description of this modulation would be sufficient.

RESPONSE: We are very happy to clarify the impact of nAChRs. We have added a few sentences to the end of the voltammetry Methods section to describe how nAChRs can modulate DA release, as well as our rationale for excluding nAChR activity in this study.

3. Aside from nAChRs, the D3 receptor has also been shown to regulate DAT activity (Zapata et al., 2007, PMID:17923483). The authors should consider potential D3 receptor involvement in their results.

RESPONSE: Indeed, the D3 receptor has been shown to regulate DAT activity in a HEK-cell based expression system (Zapata et al., 2007) and in NAc to regulate cocaine potency at inhibiting uptake by the DAT but not on mobilizing vesicle release (McGinnis et al 2016 PMID:27393374). The D3 receptors does not however appear to strongly regulate DA release in previous studies (Schmitz et al 2002). We think that probing whether D3 receptors modulate how DATs regulate short-term plasticity in DA release is beyond the scope of the present study. However, we have added a sentence and references to the Discussion on P17 to illustrate that (line 385) “DAT function is also modulated by DA D2 and D3-receptors [79,80]”, and that DAT function (line 388) could therefore be “modulated by DA”.

4. Rationale for concentrations of cocaine, methylphenidate and nomifensine should be provided.

RESPONSE: We have now clarified within the manuscript, P23 that “DA uptake inhibitors were used at standard concentrations that have detectable outcomes on uptake kinetics concentrations but are lower than those that lead to run-down of release and/or have non-selective effects as local anaesthetics or at nAChRs [46,92,93]”

5. Experiments in Supplementary Figure S4 are confusing. In the text on page 10, it is stated that “the reduction in [Ca²⁺]_o was compensated by equimolar substitution with Mg²⁺”, but the figure shows “addition” of Mg²⁺. Please clarify.

RESPONSE: We thanks the reviewer for pointing out this source of confusion. We have rephrased the text on P10 and P21 Methods to explain that “the reduction in Ca²⁺ was compensated by substitution with Mg²⁺”. We have also overhauled this Supplementary Figure and rewritten the Figure legend (previously Fig S4, now S6) to make clear that the “addition” of Mg²⁺ to which we previously referred, actually increases the concentration of Mg²⁺ in the buffer from 1.3 mM in normal conditions, to 2.5 mM to substitute for a decrease in [Ca²⁺]_o from 2.4 to 1.2 mM. We hope this is now clearer. We have also included an extra and relevant piece of data which shows what happens when Mg²⁺ concentration is increased to substitute for reducing Ca²⁺. In a 1.2 mM Ca²⁺ media, an increase in Mg²⁺ from 1.3 to 2.5 mM approximately halves DA release, adding to the case that we make in the methods section that Mg²⁺ substitutions for reductions in Ca²⁺ have additional effects beyond those due to a reduction in Ca²⁺ (**new Supplementary Figure S6D**).

The authors should also comment on the variance in P2/P1 in controls at 10 ms between data shown in Figure S4.B (*now new S6B*) and Figure 7.B Given the relatively small SEMs, it appears there are differences here that should be discussed.

RESPONSE: Different subsets of experiments can show different variance in the mean data subsets depending on the [DA]_o seen at each site or on the way the data is normalised. In these cases, the different variance does not impact on overall conclusions made. Key experimental comparisons were made within

recording sites, using data from given recordings sites before and after a manipulation of interest, so overall variance between experiments should not matter.

6. Under the surgical methods section, how long was the microinjector left in place after injection of virus? Details of how the accuracy of injection site was evaluated should be included. An image showing eYFP expression, and some quantitation of transduction efficiency should be included in supplementary materials.

RESPONSE: Following injection of the virus, the needle was left in place for 10 minutes, then retracted slowly. This information has now been added to the surgical methods section on P18. We have also added a citation to our previous use of ChR2 in DAT-Cre mice using identical methods to indicate eYFP expression (Threlfell et al 2012 PMID 22794260). We did not quantify transduction efficiency. The optogenetic approaches used here were not the main approach or focus of the study, but were used rather to corroborate our findings from electrical stimulation. We required only that striatal DA release could be driven using light stimuli.

7. It would be useful to include some representative voltammograms in either Figure 1, or in supplementary materials.

RESPONSE: Yes, we agree. We have included representative voltammograms as new insets, for electrically evoked DA release in Figure 1a, and for optogenetically evoked DA in 2g.

8. Similarly, results of calibrations of electrodes should be reported in supplementary materials. How did r^2 values compare between pre- and post-electrode calibrations?

RESPONSE: The range of electrode sensitivities to DA was approximately 5–25 nA/ μ M. This information has now been added to the methods section on P19. In these experiments, our electrodes are not routinely calibrated prior to recording. In our experience, electrodes rapidly lose some initial sensitivity after first exposure to tissue, and the post calibration is a more accurate representation of electrode sensitivity seen in tissue. Each electrode was used only for a single experiment and only the post-calibration factor was used to calculate [DA]_o.

9. Please clarify if recording sites were always targeted to the same area in CPu and NAc (pages 19-20).

RESPONSE: Recording sites in CPu were targeted to the dorsal half of the dorsolateral striatum, spanning dorsomedial and dorsolateral regions, while recordings in the nucleus accumbens core were made within approximately 200 μ m of the anterior commissure. We have clarified these details within the methods section on P21.

10. In Figure 1 caption, please indicate more clearly what the arrows are indicating. Also please clarify that where error bars are not visible they fall within the bounds of the symbol.

RESPONSE: Thanks, have done both.

11. In Figure 4, reference to the colors is confusing. It seems that in line 3 of the caption *grey* should be *red*, likewise in second to last line.

RESPONSE: Good spot – now clarified.

12. Is it possible that compensation in synapsin III knockout mice could mask a possible role for this protein on the effects of cocaine on STP? Brief discussion of this possibility and alternative approaches should be included.

RESPONSE: We agree, and have now modified the discussion of this data on P16 to state that “There might be redundancy within the synapsin family to continue to support such a synapsin-dependent mechanism in the absence of synapsin3”. Our new Ca²⁺ imaging data (described in responses to Reviewer 1) also necessitated reintroduction of this idea. We are glad to have the opportunity to do so.

Minor corrections

Page 3, 5 lines from bottom, the word “rather” should be deleted.

RESPONSE: Thanks. Now deleted.

Page 4, first line of second paragraph, “release probability (Pr)” has already been previously defined.

RESPONSE: Now deleted.

[DA]_o should be defined when first used in text.

RESPONSE: Now added.

Page 8, third line of paragraph one, “relieving” should be “relieved”.

RESPONSE: Sentence now modified.

Page 11, last line of first paragraph, “raise” should be “rise”; 5 lines from bottom, should “vesicle pool” be “vesicular pool”?

RESPONSE: Now modified.

Page 13, 12 lines from top, “...but might will alter...” should be “...but might well alter...”.

RESPONSE: Thanks. Sentence now modified.

Page 14, first line of second paragraph, delete second “together” in “Together, these findings together...”

RESPONSE: Sentence now modified.

Page 15, 10 lines from bottom, “...In any event, It is...”, i in It should be lower case.

RESPONSE: Sentence now modified.

Page 19, top line, “IPIs of 10 to 25 ms IPIs...” delete second IPI.

RESPONSE: Thanks for spotting this typo. Now deleted.

Check references for consistency in style, e.g. in most places journal names are abbreviated, but in others they are written in full.

RESPONSE: Thanks for spotting this inconsistency. We have now checked for consistency.

Why are the x-axes different in Figure 2.C and Figure 2.H? Also, it is not entirely clear why there are 6 different Ca^{2+} concentrations shown in 2.H, and only 3 in 2.C. Presumably more concentrations were tested in 2.H. This was not clearly indicated.

RESPONSE: These x-axes are linear in 2C and logarithmic in 2H, to best represent the spread of data collected. We have clarified in the fig legend that 2H is on a log scale. As the reviewer surmises, more concentrations were used in the experiments for which data are presented in 2H. These experiments interrogated the Ca^{2+} dependence of optogenetically evoked DA release, which is less well characterised than the electrical release shown in Fig 2C (for which a full data set was published previously in Brimblecombe et al 2015 PMID: 25533038), and we therefore wanted a more extensive data set.

The x-axes in Figure 3.E should be made the same.

RESPONSE: Thanks for spotting this inconsistency. We have now made these axes similar in a revised Figure 3.

Figures 4.A and 4.E are missing the vertical scale bar.

RESPONSE: We originally omitted scale bar because the data are normalised, but as requested we have now added scale bars and revised the fig legend accordingly.

Control data in Figures 6.A, C, E, G and I are missing the horizontal scale bar, and Figures 6.D, F, H and J are missing the y-axis title.

RESPONSE: The horizontal scale bars were shown once in those panels, alongside the cocaine data, and applied to all data, but to make clearer, we have separated the x and y scale bars to clarify that the x bar applies to all the data. The y-axis titles have been reinstated. Thanks for spotting.

Control data in Figures 8.A and E are missing the horizontal scale bar.

RESPONSE: As for fig 6, the horizontal scale bars were included once in those panels alongside the cocaine data, but to make clearer, we have separated the x and y scale bars and depicted one free-standing x scale bar that fits all data in that panel.

Data in Figure S4.A are missing scale bars.

RESPONSE: We originally omitted scale bar as data are normalised, but as requested we have now added scale bars and revised the fig legend accordingly.

Control data in Figure S5.A are missing horizontal scale bars.

RESPONSE: As for fig 6, the horizontal scale bars were included once in those panels alongside the cocaine data, but to make clearer, we have separated the x and y scale bars and depicted one free-standing x scale bar that fits all data in that panel.

REVIEWERS' COMMENTS:

Reviewer #1 (Remarks to the Author):

This study presents the interesting observations that 1) short-term depression of striatal dopamine release is largely release-independent and 2) that the dopamine-transporter regulates short-term plasticity. Both are convincingly shown. In my initial review, I raised concerns that the authors' references/suggestions regarding spike fidelity, axonal excitability, etc were too strong given the indirect nature their voltammetry measurements. In response to this, the authors performed new experiments of GCaMP6 population calcium measurements from axons in striatum. The experiments corroborate their previous findings and significantly improve the manuscript. Although additional questions remain regarding the mechanism of why STP and release are uncoupled, my feeling is that the initial observations along with the additional Ca imaging experiments provided are sufficient.

Reviewer #2 (Remarks to the Author):

The authors addressed properly my questions. I have no further comment.

Reviewer #3 (Remarks to the Author):

The authors are to be commended on their detailed response to the initial reviews, and their revised manuscript, which includes new data and more methodological details. This reviewer has no further concerns.

REVIEWERS' COMMENTS:

Reviewer #1 (Remarks to the Author):

This study presents the interesting observations that 1) short-term depression of striatal dopamine release is largely release-independent and 2) that the dopamine-transporter regulates short-term plasticity. Both are convincingly shown. In my initial review, I raised concerns that the authors' references/suggestions regarding spike fidelity, axonal excitability, etc were too strong given the indirect nature their voltammetry measurements. In response to this, the authors performed new experiments of GCaMP6 population calcium measurements from axons in striatum. The experiments corroborate their previous findings and significantly improve the manuscript. Although additional questions remain regarding the mechanism of why STP and release are uncoupled, my feeling is that the initial observations along with the additional Ca imaging experiments provided are sufficient.

Reviewer #2 (Remarks to the Author):

The authors addressed properly my questions. I have no further comment.

Reviewer #3 (Remarks to the Author):

The authors are to be commended on their detailed response to the initial reviews, and their revised manuscript, which includes new data and more methodological details. This reviewer has no further concerns.

AUTHORS' RESPONSE

We thank the Reviewers for their input which has significantly improved the manuscript. We are glad that the revised manuscript has now addressed their concerns.